# Multiplexed multicolor antiviral assay amenable for high-throughput research

Li-Hsin Li [1,2], Winston Chiu [1,10], Yun-An Huang[3,4,10], Madina Rasulova [5], Thomas Vercruysse [5,8], Hendrik Jan Thibaut [5], Sebastiaan ter Horst [1,9], Joana Rocha-Pereira [1], Greet Vanhoof[6], Doortje Borrenberghs[7], Olivia Goethals[7], Suzanne J. F. Kaptein [1], Pieter Leyssen [1], Johan Neyts [1] & Kai Dallmeier [1,2] ✉

To curb viral epidemics and pandemics, antiviral drugs are needed with activity against entire genera or families of viruses. Here, we develop a cell-based multiplex antiviral assay for high-throughput screening against multiple viruses at once, as demonstrated by using three distantly related ortho-flaviviruses: dengue, Japanese encephalitis and yellow fever virus. Each virus is tagged with a distinct fluorescent protein, enabling individual monitoring in cell culture through high-content imaging. Specific antisera and small-molecule inhibitors are employed to validate that multiplexing approach yields comparable inhibition profiles to single-virus infection assays. To facilitate downstream analysis, a kernel is developed to deconvolute and reduce the multidimensional quantitative data to three cartesian coordinates. The methodology is applicable to viruses from different families as exemplified by co-infections with chikungunya, parainfluenza and Bunyamwera viruses. The multiplex approach is expected to facilitate the discovery of broader-spectrum antivirals, as shown in a pilot screen of approximately 1200 drug-like small-molecules.

Phenotypic screening of large libraries of small molecules in cell-based antiviral assays has been proven as a successful strategy to identify inhibitors of viral replication. Ideally, the assays used should be sufficiently robust, stringent, and scalable to be amenable to automation for high throughput. Such approaches have led to the discovery of highly potent antiviral drugs for the treatment of important pathogens such as the human immunodeficiency virus (HIV)[1–3] and the hepatitis C virus (HCV)[4,5]. We have used such assays for the development of a highly potent dengue virus (DENV) inhibitor, which is now in clinical development[6,7]. Despite obvious advances in discovering antivirals that target a particular pathogen, in the context of epidemic and pandemic preparedness, antiviral drugs are needed to be rather active against an entire genus or even different families of viruses[8–10]. Such broader-acting antivirals can be expected to be effective if a new virus emerges within a certain virus genus or family. For example, had there been a stockpile of pre-emptively developed pan-coronavirus inhibitors available at the start of the SARS-CoV-2 pandemic, it could have had a major impact on global infection dynamics[11]. Consequently, to prepare for future pandemics, traditional monovalent drug discovery designs (testing one virus at once)[12–14] may need to be tailored towards an inherently extended target scope, enabled by methodologies that facilitate the search for such broader-acting inhibitors.

[1]KU Leuven Department of Microbiology, Immunology and Transplantation, Rega Institute, Laboratory of Virology and Chemotherapy, Leuven, Belgium. [2]Molecular Vaccinology and Vaccine Discovery group, Leuven, Belgium. [3]KU Leuven Department of Neuroscience, Research Group Neurophysiology, Laboratory for Circuit Neuroscience, Leuven, Belgium. [4]Vlaams Instituut voor Biotechnologie, Neuro-Electronics Research Flanders (NERF), Leuven, Belgium. [5]KU Leuven Department of Microbiology, Immunology and Transplantation, Rega Institute, Laboratory of Virology and Chemotherapy, Translational Platform Virology and Chemotherapy (TPVC), Leuven, Belgium. [6]Janssen Therapeutics Discovery, Janssen Pharmaceutica, NV Beerse, Belgium. [7]Janssen Global Public Health, Janssen Pharmaceutica, NV Beerse, Belgium. [8]Present address: AstriVax, Heverlee, Belgium. [9]Present address: Cerba Research, Rotterdam, The Netherlands. [10]These authors contributed equally: Winston Chiu, Yun-An Huang. ✉e-mail: kai.dallmeier@kuleuven.be

Here, we describe a cell-based multiplex multicolor antiviral assay that allows to profile antiviral activities concomitantly against multiple viruses. For example, we choose three distantly related orthoflaviviruses[15], i.e., the dengue virus (serotype 2, DENV-2), Japanese encephalitis virus (JEV) and yellow fever virus (YFV). For these viruses, antiviral treatments are currently not available. Same applies to other medically important members of the Orthoflavivirus genus such as the West Nile virus (WNV), Zika virus (ZIKV), tick-borne encephalitis virus (TBEV)[16–18] and obviously yet unknown orthoflaviviruses that may emerge in the future. Whereas classical cell-based approaches, which are used to assess antiviral effects of compound libraries, rely on the quantification of the reduction of either (i) a virus-induced CPE (cytopathic effect)[12] or (ii) of viral RNA yields (by qRT-PCR)[19,20], our image-based screening assay uses reporter viruses and real-time quantification of infection by means of high-content imaging (HCI). Our assay can be performed in a convenient homogenous format, it does not require fixation, immunostaining nor molecular analysis and is hence amenable to high-throughput screening (HTS). By multiplexing several live reporter viruses, each tagged with spectrally distinct fluorescent proteins (FP), individual differences in drug sensitivity can be visualized in parallel and directly compared. The current set up uses reporter viruses, including DENV-2/mAzurite (blue), JEV/eGFP (green), and YFV/mCherry (red), in combination with Vero cells expressing near-infrared FP (V-NIR cells) as a common substrate. Likewise, possible side-effects of treatment (cytotoxicity) are directly quantified by the reduction in the number of V-NIR cells. To expand the approach beyond orthoflaviviruses, we also set up multiplex assay conditions using combinations of chikungunya, parainfluenza and Bunyamwera reporter viruses, representing different virus families. To simplify the representation of the specificity, potency, and broad-spectrum coverage of hit compounds emerging from compound library screens, we have developed a kernel that automatically extracts and converts multidimensional HTS data, representing dose-dependent antiviral activities regarding multiple viruses, into a simple RGB (Red-Green-Blue) color code. The multiplex broad-spectrum anti-Orthoflavivirus assay pipeline has been validated using a panel of reference sera and small-molecule inhibitors with known activity and specificity. This validation is conducted within an automated combined robotics-biosafety containment system (pathogen-in/operator-out)[21,22]. The feasibility and robustness of our approach is demonstrated through a pilot-scale screen of 1256 drug-like small-molecules, revealing 49 hits, of which 11 with dual activity and one with activity against all three orthoflaviviruses.

## Results

### Multiplexed multicolor anti-orthoflavivirus assays

We and others have used genetically tagged reporter viruses, e.g., DENV-2 tagged with a red-fluorescent mCherry reporter[23] for antiviral studies. In principle, this approach should be extendable by combining viruses that express different spectral variants of GFP and its orthologs[24,25]. Here, we demonstrate the principle and flexibility of such multiplex/multicolor approach by using a DENV-2 strain to generate five different recombinant viruses, each expressing an array of five different FPs (Fig. 1a, b and Supplementary Methods; Supplementary Fig. 1). All five viruses, DENV-2/mAzurite (blue), DENV-2/eGFP (green), DENV-2/mCitrine (yellow), DENV-2/mCherry (red) and DENV-2/mMaroon (dark red) remained fully replication competent (Supplementary Fig. 2). The trade-off in replication resulting from the insertion of any FP was minimal, yielding viruses with comparable fitness, allowing multiple infections in any combination[26]. However, DENV-2/eGFP, DENV-2/mCitrine and DENV-2/mMaroon appeared less suitable for image-based analysis due to spectral overlap (eGFP and mCitrine readout) or low brightness of the reporter (mMaroon), respectively (Supplementary Fig. 1).

Next, we wanted to establish a multiplexed anti-orthoflavivirus assay by simultaneously infecting V-NIR cells with a mixture of at least three distinct orthoflaviviruses. Considering that each virus has its own replication kinetics and host cell interactions, mutual interference may pose a major challenge. For conceptual proof, we selected DENV-2, JEV, and YFV as three clinically relevant mosquito-borne orthoflaviviruses that are sufficiently phylogenetically distant. These should be representative for the genus Orthoflavivirus, making them suitable for establishing a high-throughput antiviral screen for molecules with broader-spectrum anti-orthoflavivirus activity. To facilitate individual scoring, each virus was tagged with a spectrally distinct reporter (Fig. 1c, d). These viruses retained replication competent as demonstrated by HCI of their multiplicity of infection (MOI) -dependent growth kinetics (Fig. 1e) and their ability to form plaques on BHK21J cells (Fig. 1f). Similar to the evaluation done for the DENV-2 reporter viruses[23] (Supplementary Figs 1 and 2), we compared the phenotype and replication fitness of YF17D/mCherry and JEV/eGFP to that of parental YF17D and JEV SA14-14-2 viruses (Supplementary Fig. 3). Smaller plaques caused by the reporter viruses suggest some degree of attenuation. Nonetheless, their respective growth kinetics showed no significant differences compared to their parental strains, confirming their full replication capacity.

For an optimal combination of fluorophores, we relied on our experience with the five DENV-2 reporter viruses, assuming the actual choice of FP hardly influenced viral replication characteristics (Supplementary Figs. 1 and 2). We hence selected fluorophores that provided optimal spectral separation (Supplementary Fig. 1 and Supplementary Table 1) within the three reporter orthoflaviviruses, namely DENV-2/mAzurite (blue), JEV/eGFP (green) and YF17D/mCherry (red). These three reporter viruses and V-NIR cells (Fig. 1d) were used to develop a 4-color multiplexed co-infection model (Fig. 2).

Unique infection rates were observed for each of the three different viruses (Fig. 1e; Supplementary Figs. 2 and 3), thus requiring optimization of co-infections parameters. To that end, viral growth kinetics were compared under two conditions: V-NIR cells infected by (i) only one of the viruses (Fig. 1e) versus those infected by (ii) virus mixtures containing equal infectious titers of each virus (Fig. 1g). In the single-virus infection assay, YF17D/mCherry was shown to grow most efficiently, followed by JEV/eGFP, and DENV-2/mAzurite had the slowest replication kinetics. Also, replication kinetics of JEV/eGFP had slowed down after 5 days post infection (dpi), whereas this was not the case for DENV-2/mAzurite (Fig. 1e). In line with these individual viral growth characteristics, the majority of the infected cells expressed JEV/eGFP within the first 2 days of the multiplexed infections (Fig. 1g). Later, YF17D/mCherry became dominant. Compared to JEV/eGFP and YF17D/mCherry, DENV-2/mAzurite grew much slower and thus accounted for a minority of the infected cells under all tested conditions. Notably, all three viruses could be individually tracked. Interestingly and importantly, V-NIR cells co-infected by more than one reporter virus at once were barely detectable. This observation corresponds to superinfection exclusion which is commonly seen among viruses in the Orthoflavivirus genus[26,27], the larger *Flaviviridae* family[28,29] and beyond[30–32].

Based on these findings, the ratio of DENV-2/mAzurite, JEV/eGFP, and YF17D/mCherry needed to be adjusted towards a more balanced infection ratio for all three reporter viruses, preferably at each time point and certainly at endpoint. Therefore, the number of cells initially infected by YF17D/mCherry had to be kept relatively low, and the time to readout limited. A readout at 3 or 4 dpi was expected to be favorable[23] since virus-induced CPE may not yet have a major impact on viable cell counts (Fig. 1e; Supplementary Fig. 2c).

Following this rationale, we tested five different ratios (R1 to R5) in the multiplexed virus infection experiment (Fig. 2a) and monitored the dynamics of replication up to 7 dpi. Finally, R5 was selected as the most appropriate mixture ratio, considering the comparable percentages of

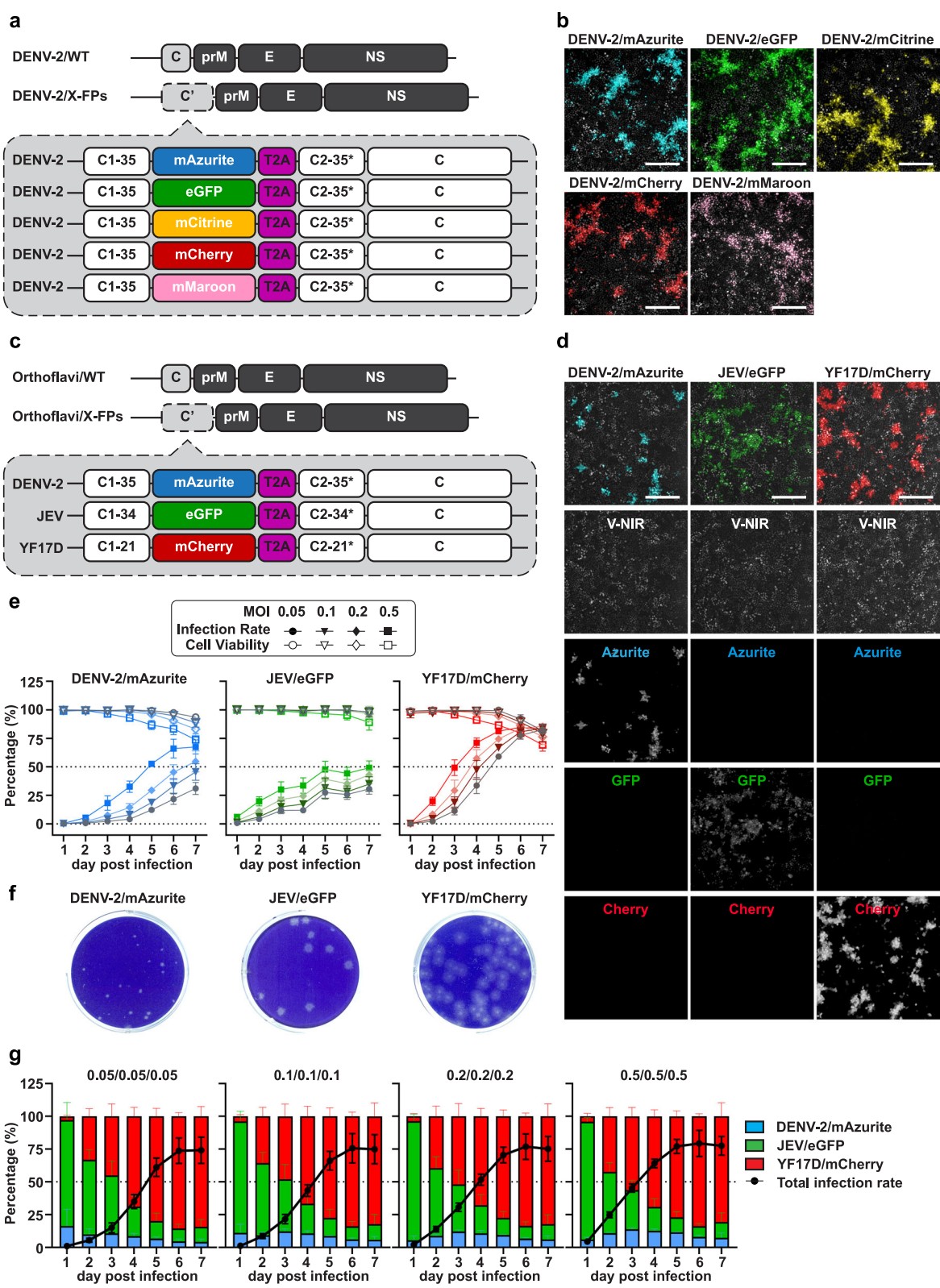

infected cells for each of the three viruses and this consistently resulted in high Z' values for all three viruses. 3 dpi was then determined as endpoint for further multiplexed assays (Fig. 2b).

## Assay validation using specific antisera and antiviral reference compounds

To validate the multiplex assay, the potency and specificity of a series of reference compounds with known anti-flavivirus activity was assessed. To this end, we used (i) antisera containing virus-specific neutralizing antibodies (Fig. 3a–c); (ii) two previously validated pan-flavivirus inhibitors, interferon alpha (IFN-α) and NITD008[33], each with a different molecular mechanism of action; and (iii) our recently reported dengue-specific inhibitor, JNJ-A07[6,7] (Fig. 3d, f). Both single-virus assays (Supplementary Fig. 4) and multiplex-virus assays (Fig. 3) were performed in parallel. To that end, V-NIR cells were infected with DENV-2/mAzurite (0.5 MOI), JEV/eGFP (0.5 MOI), and YF17D/mCherry

**Fig. 1 | Comparative characteristics of reporter orthoflaviviruses. a** Design of DENV-2 reporter viruses[23]. Individual reporter viruses are identical except for their respective fluorescent proteins (FPs) inserted as reporters. **b** HCI readout of cells infected by DENV-2 reporter viruses. Vero cells expressing near-infrared fluorescent protein (V-NIR) infected with respective DENV-2/X-FPs at an MOI of 0.1. Images taken on live cells at 5 days post-infection (dpi). Scale bar: 255 μm. **c** Design of DENV-2/mAzurite, JEV/eGFP and YF17D/mCherry. Reporter genes flanked by a repeat for the first 21-35 amino acids of the respective capsid protein (C2-35* for DENV-2, C2-34* for JEV, and C2-21* for YF17D) and a ribosome-skipping T2A peptide. **d** V-NIR infected with DENV-2/mAzurite, JEV/eGFP or YF17D/mCherry. Representative fluorescent images taken by HCI at 3 dpi with an MOI of 0.2. Scale bar: 255 μm. **e** Kinetics of individual reporter flavivirus infections. V-NIR individually infected by either DENV-2/mAzurite, JEV/eGFP or YF17D/mCherry at an MOI of 0.05, 0.1, 0.2 or 0.5. Fluorescent signals detected daily until 7 dpi by HCI. Cell viability defined as number of NIR positive cells in virus infected wells versus number in the uninfected wells; virus infectivity defined as number of cells with respective fluorescent signals versus number of NIR positive cells in the same well. **f** Plaque phenotype of DENV-2/mAzurite, JEV/eGFP and YF17D/mCherry. Representative photographs taken from titration of working virus stocks at a dilution of 1:10⁶, 1:10⁷ and 1:10⁵, respectively (see Supplementary Figs 2 and 3 for comparison with parental viruses). **g** Dynamics of mixed orthoflaviviruses infections. V-NIR infected with mixtures of DENV-2/mAzurite, JEV/eGFP and YF17D/mCherry at equal ratios (1:1:1); at an MOI of 0.05, 0.1, 0.2 or 0.5. NIR and virus-specific fluorescent signals detected by HCI daily until 7 dpi. Total infection rate defined as total number of cells with Azurite, GFP or Cherry signals versus number of NIR positive cells in the same well (black line). Individual columns show the ratios of DENV-2/mAzurite, JEV/eGFP and YF17D/mCherry among all infected cells in the same well at each timepoint. Data represent means ± SD of $n = 3$ independent experiments. Source data are provided as a Source Data file.

(0.2 MOI) separately (Supplementary Fig. 4), or a mixture thereof each containing the same amount of virus as in the respective single-virus assays (Fig. 3).

Serum neutralizing tests (SNT) performed (in parallel) by single-virus assays (Supplementary Fig. 4a) or multiplex-virus assays (Fig. 3a) yielded very similar $SNT_{50}$ and $SNT_{90}$ values (Table 1). Obviously, a dose-dependent change in the proportion of infected cells was observed in the multiplex assay (Fig. 3a and Supplementary Figs 5–7). Those viruses that are not neutralized remain detectable, and the pie charts illustrate how fluorescent signals disappear from the mixed population of viruses, indicating which viruses are sensitive to the neutralization (Fig. 3b). Interestingly, when a particular virus was neutralized, the other two viruses replicated more efficiently as compared to the untreated conditions: a likely consequence of reduced competition (Fig. 3a). Nonetheless, data obtained by either single-virus or multiplex-virus assay correlated well ($r = 0.98$, Pearson test). Bland-Altman analysis confirmed this, and data obtained by multiple-virus SNT are robust and lie within an acceptable error range as compared to the single-virus SNT (within limits of agreement, i.e., bias ±95% CI of mean differences). Limits of agreement (LOA) were found between single-virus SNT and multiple-virus SNT values, with an average bias of −4 (LOA from −22 to 14) (Fig. 3c).

When inhibitors with known anti-flavivirus activity were assessed using in parallel a single-virus (Supplementary Fig. 4b) and a multiplex-virus set-up (Fig. 3d), both assays yielded overall similar inhibition profiles. Whereas the pan-flavivirus inhibitors, IFN-α and NITD008, suppressed replication of all three viruses, the dengue-specific inhibitor JNJ-A07 exclusively ablated the replication of DENV-2/Azurite in a dose-dependent manner (Fig. 3d, e). $EC_{50}$ values obtained in either single-virus or multiplex-virus assay were almost identical for the directly acting antivirals NITD008 (targeting the flavivirus polymerase)[33] and JNJ-A07 (targeting the dengue NS3/NS4B interaction)[6,7]; respectively in the low micromolar and sub-nanomolar range (Table 2).

Inhibition by IFN-α (host-cell targeting) was more pronounced for JEV/eGFP and YF17D/mCherry ($EC_{50}$ ~ 2 IU/mL in either assay) than for DENV-2/mAzurite ($EC_{50}$ ~ 9 IU/mL in single-virus assay). This biological difference was further amplified in the multiplex assay ($EC_{50}$ ~ 42 IU/mL for DENV-2), which is in line with an increase in relative replication fitness of DENV-2 conferred by early blocking of the other viruses. Due to this competitive advantage of DENV-2 over JEV and YFV in presence of IFN-α, the overall correlation between antiviral assays (Fig. 3f and Supplementary Fig. 8) appeared slightly lower (Pearson coefficient $r = 0.94$; coefficient of determinations $R^2 = 0.88$) than that observed for the SNTs (Fig. 3c). Nevertheless, overall single-virus and multiplex assays correlated well, within limits of agreement in Bland–Altman analysis (Fig. 3f and Supplementary Fig. 8). Limits of agreement (LOA) were found between single-virus antiviral assay and multiple-virus antiviral values, with an average bias of −5 (LOA from −33 to 23).

## Multiplex infection with viruses from different families

To help the identification of compounds with an even broader antiviral spectrum against viruses that are more distantly related, our basic assay setup needs to be modified to include representatives from different virus families in HTS campaigns. For proof of concept, we established two other triple multiplex infections; combining DENV-2/mAzurite with parainfluenza virus 1 (PIV1 tagged with eGFP, PIV1/eGFP)[34,35] and either chikungunya virus (CHIKV tagged with mCherry, CHIKV/mCherry; Fig. 4a–c)[36,37] or Bunyamwera virus (BUNV tagged with mCherry, BUNV/mCherry; Fig. 4d–f)[38,39]. Careful titration by using different MOIs was required to balance the varying infection kinetics (Fig. 4b, e), while avoiding loss of cells by CPE (Fig. 4c, f). This proved most challenging for combinations containing the rapidly replicating BUNV/mCherry (Fig. 4e). We used Vero cells as the preferred cell substrate here due to their general high susceptibility to viruses[40]. However, multiplexing was also possible with cells of human origin such as A549 (lung carcinoma) (Fig. 4g–i) and Huh-7 (hepatoma) cells (Supplementary Fig. 9).

## RGB model to visualize and deconvolute multidimensional quantitative datasets

Our multiplexed reporter virus assay delivers complex multidimensional data, regarding specificity and potency of an antiviral compound. Data may vary over several orders of magnitude depending on the viruses used. To readily visualize such differences and accelerate data analysis, we developed an intuitive kernel that is based on the RGB color model corresponding to the three fluoroproteins [mCherry; red (R), eGFP; green (G), and mAzurite; blue (B)] used as reporters (Fig. 5a). In the RGB model, these three primary colors of light are added in various combinations and relative intensities (range: 0–100%) to produce a particular mixed color (Fig. 5a, b). The resulting color can likewise be defined by the respective (R, G, B) coordinates and hence expressed as a distinct point in a 3-dimensional (3D) space (Fig. 5c and Supplementary Movie 1). If all components of the RGB triplet are at 0% (0,0,0), corresponding to full inhibition of all three reporter viruses, the result is black; if all are at 100% (100,100,100), corresponding to no antiviral activity towards any of the tested viruses, the result is completely white (Fig. 5c).

We took advantage of this RGB paradigm to convert specific antiviral activities for each individual virus and condition tested in our multicolor assay into a simple color code. More specifically, under conditions of unrestricted virus growth, mixing colors for YF17D/mCherry (R), JEV/eGFP (G) and DENV-2/mAzurite (B) results in "white" (Fig. 5b, top panel as in virus control, VC). By contrast, full inhibition of all three viruses will result in "black" (as cell control, CC). For instance, treatment with an extremely potent DENV-specific antiviral will result in 100% inhibition of only the "blue" reporter virus (DENV-2/mAzurite) and no inhibition of the red (YF17D/mCherry) and green (JEV/eGFP)

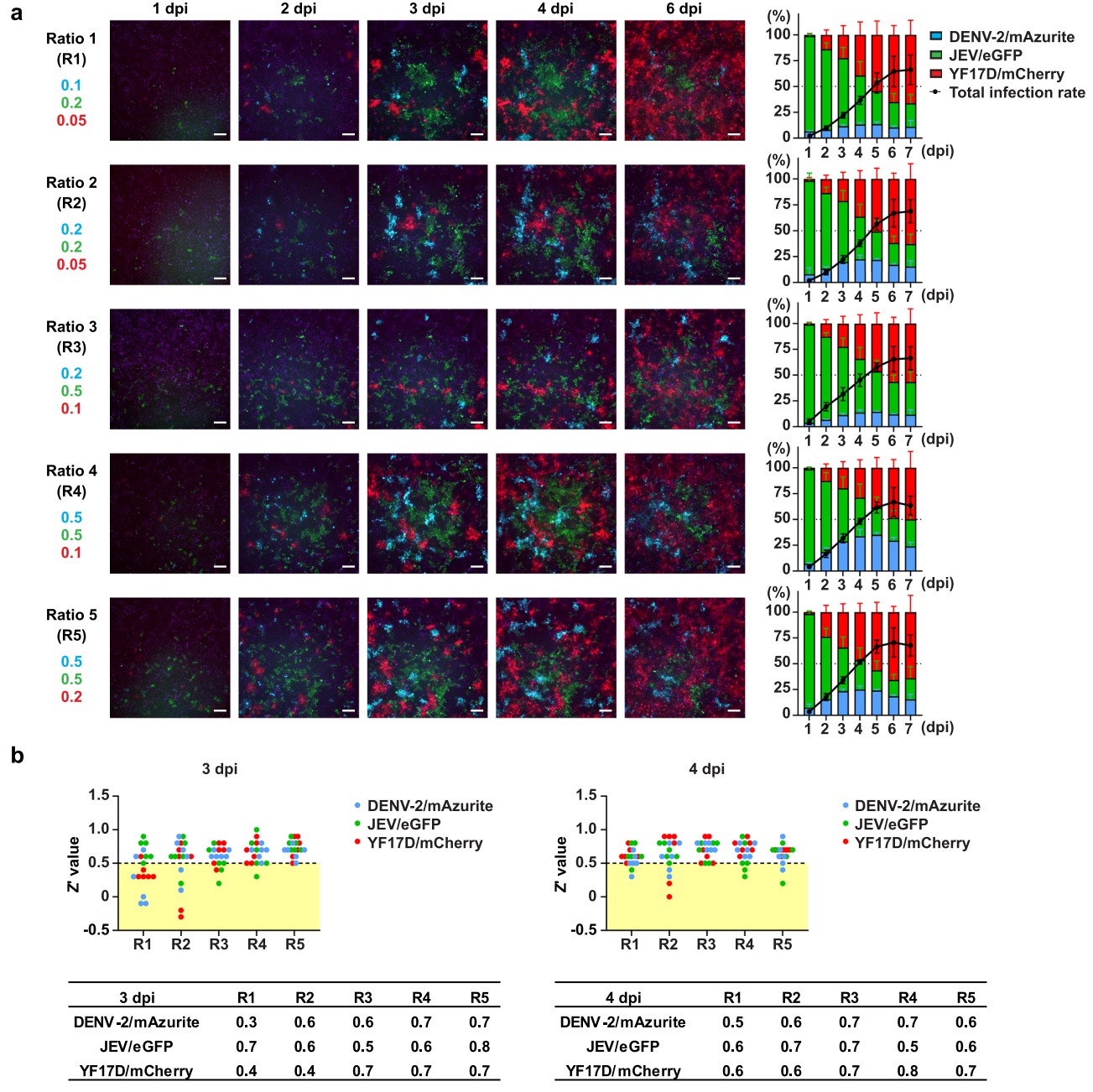

**Fig. 2 | Optimization of multiplex Orthoflavivirus infections. a** Testing of multiple ratios of infection. Representative images of V-NIR cells infected with DENV-2/mAzurite, JEV/eGFP and YF17D/mCherry altogether in five different ratios (R1-R5). Fluorescent signals of NIR, mAzurite, eGFP and mCherry were detected by HCI at 1, 2, 3, 4, and 6 dpi. Scale bar: 100 μm. Total infection rates are defined as total number of cells with Azurite, GFP or Cherry signals versus number of cells with NIR signals in the same well. Columns show the ratio of cells infected with DENV-2/mAzurite, JEV/eGFP or YF17D/mCherry out of total infected V-NIR cells in the same well at each time point. The data presented means ± SD of $n = 3$ independent experiments. **b** Assay robustness. Individual Z' values for detection of DENV-2/mAzurite, JEV/eGFP and YF17D/mCherry were calculated for each ratio of the multiple Orthoflavivirus infections at 3 dpi (left) or 4 dpi (right). Z' values < 0.5 that are too low for HTS are shown in the yellow zone. Data collected from $n = 6$ independent experiments; values shown as mean for each ratio. Source data are provided as a Source Data file.

signal, resulting in the mixed color code "yellow" (Fig. 5b, bottom panel). Any resulting mixed color thus serves as a unique identifier of the broad-spectrum inhibition profile tied to a given antiviral or antibody. To illustrate this, a highly potent and specific DENV-2 inhibitor is represented as a yellow point in the 3D-cube (Fig. 5b, c; yellow circle; see also Supplementary Movie 2). Molecules inhibiting all three viruses equally fall on a straight diagonal line connecting coordinates (100,100,100; low potency−no inhibition) and (0, 0, 0; high potency−full inhibition), or "white" to "black" (WB) on a linear intensity scale (Fig. 5c; triangles).

Antiviral screening aims at identifying compounds with high potency. For ease of scoring and classification of promising hits that may emerge from such multiplexed screens, the 3D-cube can be further sectioned diagonally into five parallel stacks (color palettes, Fig. 5d), which are perpendicular to the WB scale (see Supplementary Movie 3 for animated projection). Hence, individual inhibitory activities can be projected to the WB scale for overall potency, from low to high (Fig. 5e; left to right). Obviously, most potent antiviral candidates with broad-spectrum activity (Fig. 5d, utmost right panel) are expected in the lower sections (in plane with or close to black triangle in Fig. 5c).

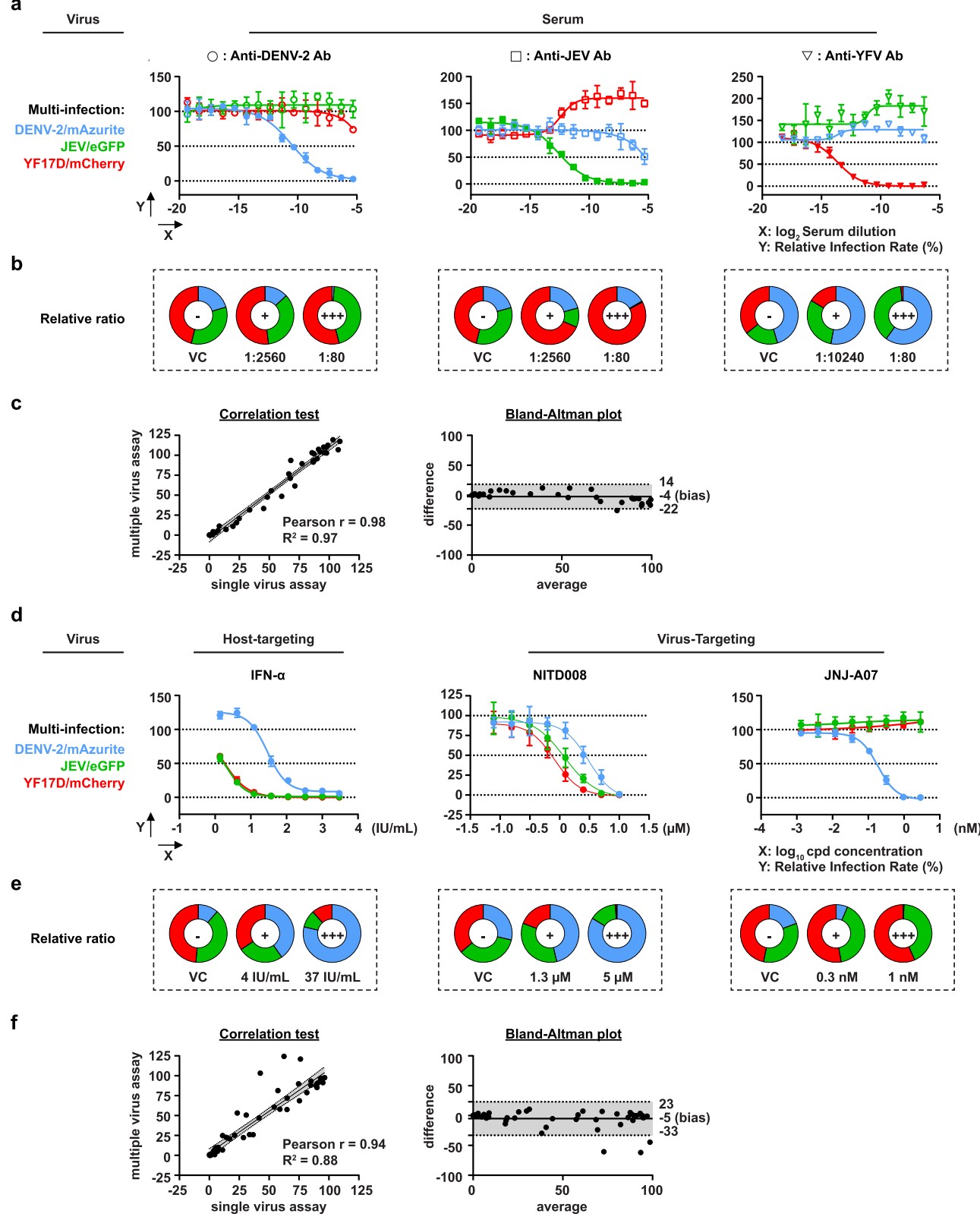

Data of inhibitors with equimolar activity against all three viruses can be readily plotted on the top of this scalebar (Fig. 5e). However, most inhibitors identified in primary antiviral screens will likely not show equal potency against all three viruses tested. Such data points are positioned towards the top of the WB scale bar according to their offset and distance from the ideal diagonal. By this means, individual values emerging from HTS of large compound libraries can easily be interpreted and compared in a 3D and 2D space.

For instance, the activity of a small molecule antiviral or antiserum resulting in full inhibition of DENV-2/mAzurite (100, 100, 0) (yellow circle in Fig. 5c; see also Supplementary Movie 2) will score in the same overall activity range as a compound with equal/comparable inhibition

**Fig. 3 | Validation of multiplex Orthoflavivirus infection assay for antiviral studies. a** Validation using sera containing virus-specific antibodies (Ab). V-NIR cells were infected with virus mixture R5 (Fig. 2), preincubated with serial dilutions of sera containing anti-DENV-2 Ab (circle), anti-JEV Ab (square), or anti-YFV Ab (triangle), respectively, as described in Methods section (SNT). Representative neutralizing curves plotted using nonlinear regression. The data presented means ± SD of $n = 3$ independent experiments. **b** Pie charts showing the percentages of DENV-2-, JEV- and YFV-infected cells versus the total infected cells at indicated dilutions. **c** Correlation analysis of single-virus SNT (Supplementary Fig. 4) and multiplex-virus SNT. The correlation between the single-virus assay and multiple-virus assay is indicated by a black solid line, with the 95% confidence intervals indicated by gray shaded areas within the dashed lines. Next to Pearson correlation (left panel), the Bland–Altman test (right panel) was used to estimate the degree of agreement between assays, comparing differences in infection rates obtained by single-virus assay (Supplementary Fig. 4) and multiplex-virus assay

with their respective average values. Dashed lines represent bias and 95% confidence intervals (lower and upper limits of agreements). **d** Validation using known Orthoflavivirus inhibitors. V-NIR cells infected with virus mixture R5 were treated with a dilution series of IFN-α [a universal host-cell targeting (orthoflavi)virus inhibitor], NITD008[33], or JNJ-A07[6,7]. Representative dose-response curves plotted using nonlinear regression. The data presented means ± SD of $n = 3$ independent experiments. **e** Pie charts showing the percentages of DENV-2-infected, JEV-infected, and YFV-infected cells versus the total infected cells at indicated concentration of inhibitors. **f** Correlation analysis of single-virus antiviral assay (Supplementary Fig. 4) and multiplex-virus antiviral assay. Pearson correlation analysis (left) and Bland–Altman test (right) were used to estimate the degree of agreement between assays as in panel (**c**). IU/mL International Units/mL, cpd compound, pie charts: − untreated virus control (VC), + representative medium dilution/medium concentration, +++ representative low dilution/high concentration. Source data are provided as a Source Data file.

**Table 1 | SNT$_{50}$ and SNT$_{90}$ values determined by single-virus or multiplex-virus assays for antisera against DENV-2, JEV or YFV**

| Infection | Virus | Anti-DENV-2 Serum | | Anti-JEV Serum | | Anti-YFV Serum | |
|---|---|---|---|---|---|---|---|
| | | SNT$_{50}$ | SNT$_{90}$ | SNT$_{50}$ | SNT$_{90}$ | SNT$_{50}$ | SNT$_{90}$ |
| Single-virus infection | DENV-2/mAzurite | 725 ± 136 | 184 ± 23 | 65 ± 4 | n.d. | 199 ± 97 | n.d. |
| | JEV/eGFP | n.d. | n.d. | 4885 ± 914 | 732 ± 75 | n.d. | n.d. |
| | YF17D/mCherry | n.d. | n.d. | n.d. | n.d. | 12759 ± 6468 | 1888 ± 157 |
| Multi-virus infection[a] | DENV-2/mAzurite | 1370 ± 70 | 168 ± 55 | 49[b] | n.d. | n.d. | n.d. |
| | JEV/eGFP | n.d. | n.d. | 4354 ± 195 | 1121 ± 21 | n.d. | n.d. |
| | YF17D/mCherry | n.d. | n.d. | n.d. | n.d. | 14444 ± 4648 | 2849 ± 393 |

Values shown as mean ± SD.

*n.d.* non-detectable.

[a]Multi-infection: Infection of DENV-2/mAzurite, JEV/eGFP and YF17D/mCherry.

[b]Only two times out of the three individual experiment reach 50% of the neutralization, with as calculated SNT$_{50}$ 1:49.

**Table 2 | EC$_{50}$ and EC$_{90}$ values for interferon alpha (IFN-α), the pan-flavivirus polymerase inhibitor NITD008, and the DENV-specific inhibitor JNJ-A07 as determined in the single-virus or multiplex-virus assays**

| Infection | Virus | IFN-α (IU/mL) | | NITD008 (μM) | | JNJ-A07 (nM) | |
|---|---|---|---|---|---|---|---|
| | | EC$_{50}$[a] | EC$_{90}$[a] | EC$_{50}$ | EC$_{90}$ | EC$_{50}$ | EC$_{90}$ |
| Single-virus infection | DENV-2/mAzurite | 8.6 ± 1.9 | 156 ± 24 | 1.5 ± 0.2 | 6.1 ± 1.3 | 0.2 ± 0.1 | 0.7 ± 0.2 |
| | JEV/eGFP | 1.9 ± 0.3 | 10 ± 1.1 | 0.9 ± 0.1 | 3.6 ± 0.4 | n.i. | n.i. |
| | YF17D/mCherry | 1.6 ± 0.1 | 9.6 ± 0.3 | 0.9 ± 0.1 | 2.7 ± 0.3 | n.i. | n.i. |
| Multi-virus infection[b] | DENV-2/mAzurite | 42 ± 9.0 | 457 ± 258 | 2.6 ± 0.7 | 7.2 ± 0.8 | 0.2 ± 0.02 | 0.7 ± 0.1 |
| | JEV/eGFP | 1.8 ± 0.3 | 9.1 ± 0.5 | 1.1 ± 0.4 | 3.7 ± 0.4 | n.i. | n.i. |
| | YF17D/mCherry | 1.9 ± 0.2 | 11 ± 1.7 | 0.7 ± 0.2 | 2.2 ± 0.3 | n.i. | n.i. |

Values shown as mean ± SD.

*n.i.* no inhibition.

[a]Extrapolated values in case lowest concentration of IFN-α used still resulted in some virus inhibition (infected cell count <100% of untreated virus control).

[b]Multi-infection: Infection of DENV-2/mAzurite, JEV/eGFP and YF17D/mCherry.

(33%) for all three viruses (66, 66, 66), or yet another compound that would inhibit YF17D/mCherry and JEV/eGFP by 50% but with no activity against DENV-2/mAzurite (50, 50, 100) (Fig. 5e). By combining the information from both figures, the profile of a treatment can be fully characterized, including its specificity (by color palette; Fig. 5d) and potency (by WB scale; Fig. 5e). Representative experimental data from multiplexed SNT assays (-SNT$_{50}$, Fig. 3a and Table 1) and antiviral assays (-EC$_{90}$, Fig. 3d and Table 2) are presented on the RGB palette and WB scale (Supplementary Fig. 10).

**Multiplex antiviral screen against orthoflaviviruses**

For proof of concept, we employed our multicolor Orthoflavivirus assay to conduct a screening of a medium-size (~1200) library of drug-like small-molecule compounds (see below) in an operator-blinded

manner. A comprehensive panel of reference compounds with diverse molecular mechanisms of action was incorporated as benchmark, comprising (i) the previously validated DENV-2-specific inhibitor JNJ-A07, (ii) the pan-flavivirus inhibitor NITD008, (iii) the broad-spectrum (+) RNA virus inhibitor 7DMA[41], (iv) the YFV NS4B inhibitor BDAA[42], and (v) the DENV capsid targeting ST-148[43]. Each reference compound was tested in 3-fold serial dilutions, repeated 5 times, to serve as quality control throughout the screening (Fig. 6a–c). As expected JNJ-A07 and BDAA resulted in a pronounced single-target activity against DENV-2/mAzurite and YF17D/mCherry, respectively (Fig. 6c). ST-148 inhibited DENV-2 replication, even though not potently, with around 50% inhibition before cytotoxicity started to dominate. NITD008 and 7DMA effectively ablated replication of all three viruses (Fig. 6c).

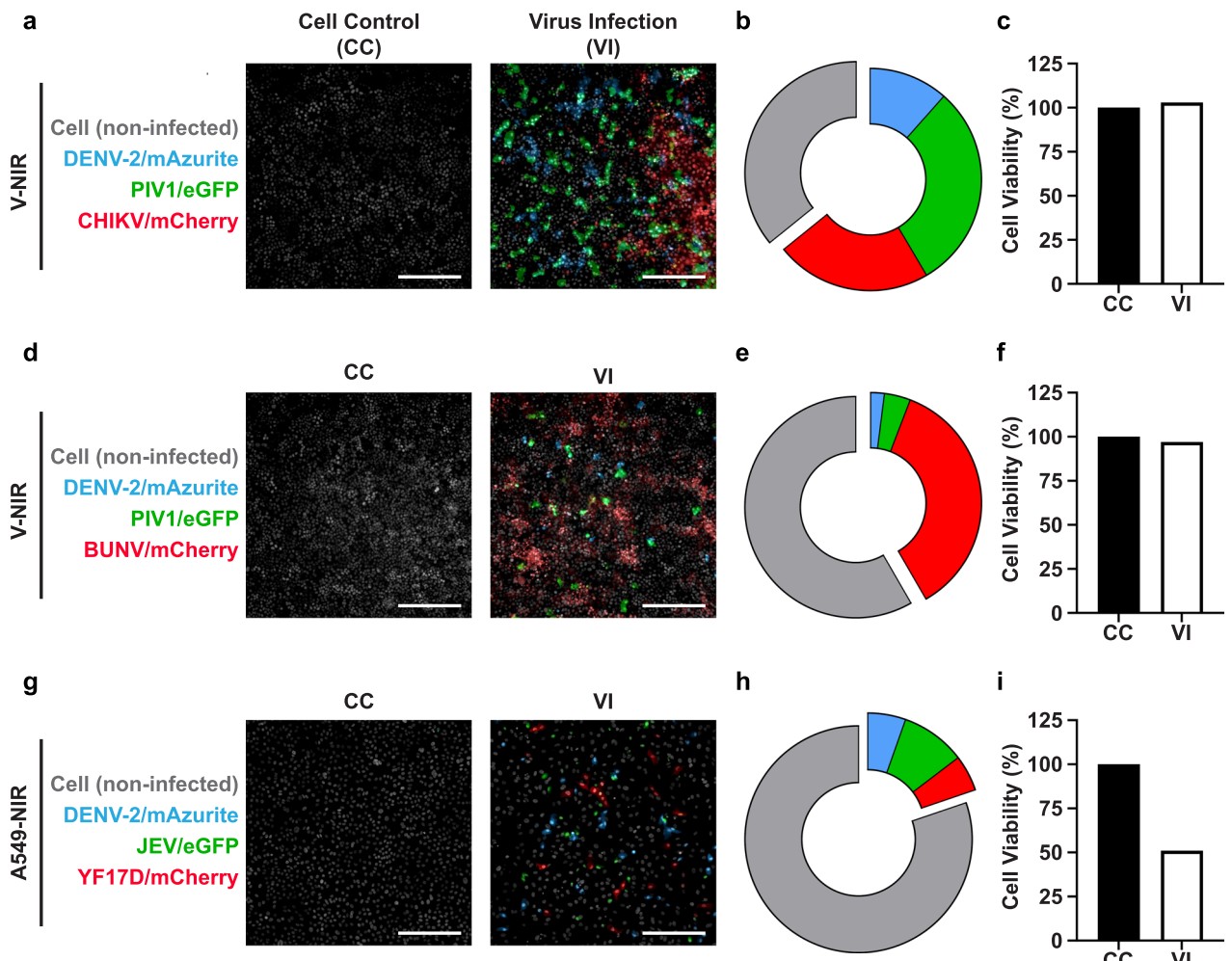

**Fig. 4 | Multiplex infections in V-NIR or A549-NIR using different triple virus combinations. a** V-NIR cells infected with a virus mixture containing DENV-2/mAzurite (0.3 MOI), PIV1/eGFP (0.3 MOI), and CHIKV/mCherry ($3 \times 10^{-5}$ MOI) for 3 days. **b** Pie chart shows the percentages of DENV-2/mAzurite, PIV1/eGFP and CHIKV/mCherry infected cells at 3 dpi as in (**a**). **c** Cell viability of the virus infected (VI) cells compared to uninfected cell controls (CC) as in (**a**). **d** V-NIR cells infected with a virus mixture containing DENV-2/mAzurite (0.3 MOI), PIV1/eGFP (0.1 MOI), and BUNV/mCherry (0.1 MOI) for 3 days. **e, f** Percentages of DENV-2/mAzurite, PIV1/eGFP and BUNV/mCherry infected cells and cell viability of VI compared to CC at 3 dpi as in

(**d**). **g** A549-NIR cells infected with a virus mixture containing DENV-2/mAzurite (5 MOI), JEV/eGFP (20 MOI), and YF17D/mCherry (6 MOI) for 3 days. **h, i** Percentages of DENV-2/mAzurite, JEV/eGFP and YF17D/mCherry infected cells and cell viability of VI compared to CC at 3 dpi as in (**g**). The cells expressing specified fluorescence signals (infected cells) are detected by HCI (Arrayscan XTI). The microscopy images shown in (**a, d, g**) are representative for each triple-virus combination. Data representative of two independent experiments except for the combination of DENV-2/mAzurite, PIV1/eGFP, and CHIKV/mCherry, which was performed only once. Scale bar: 255 μm. Source data are provided as a Source Data file.

For the pilot-scale screen, a panel of 1256 test compounds were selected from a proprietary library of small molecules with drug-like properties to represent a high chemical diversity (provided by D.B. and G.V.); supplied as DMSO stocks dispensed in 96-well plates (final concentration range 1–10 μM). This library was subjected to antiviral assessment on the CapsIt-Platform (by L.-H. L. and W.C.). The resulting HCI data were extracted, computed, and visualized using our RGB algorithm (Fig. 6d). After excluding samples with (i) pronounced cytotoxicity (reduction of primary cell counts by more than 30%; Toxicity in Fig. 6e) as primary filter, and (ii) low potency (failure to reduce infection rates by at least 70%; Efficacy Fig. 6e) as second filter, 49 hits remained. Among these, 11 exhibited dual activity and one inhibited the replication of all three orthoflaviviruses (Fig. 6f).

## Discussion

Each emergence of a new virus such as SARS (in 2004), Zika virus (in 2015/16) or SARS-CoV-2 (in 2019/2020) sparks a wave of desperate efforts to search for existing drugs that could be repurposed and used off-label for treatment of the infection. These include drugs with

questionable safety profiles[44,45] as discussed before[46,47]. For epidemic and pandemic preparedness, the development of highly potent and safe, orally available antivirals that ideally exert activity against an entire group (genus or family) of viruses will be essential. Such drugs can also be used for the treatment of viral infections for which today there is no treatment available. Drug discovery programs tend to pursue a classical one-virus-at-a-time approach. Here, we show an alternative multiplex approach using an array of reporter orthoflaviviruses to identify hits that may be suited for further development towards pan-flavivirus inhibitors in high-throughput screens (HTS). For proof of concept, we combined three reporter orthoflaviviruses, namely DENV-2/mAzurite, JEV/eGFP, and YF17D/mCherry, to develop an image-based multiplexed cell-based/phenotypic antiviral assay. Firstly, we demonstrate that such assay can be built on co-infection of a mixed population of distinct viruses in the same cell line; without losing sensitivity or specificity when viral interference is limited by carefully balancing the viral input and selecting the most appropriate endpoint.

It is tempting to speculate how far multiplexing with HCI as readout could be developed. Initially, we limited the study to the use of

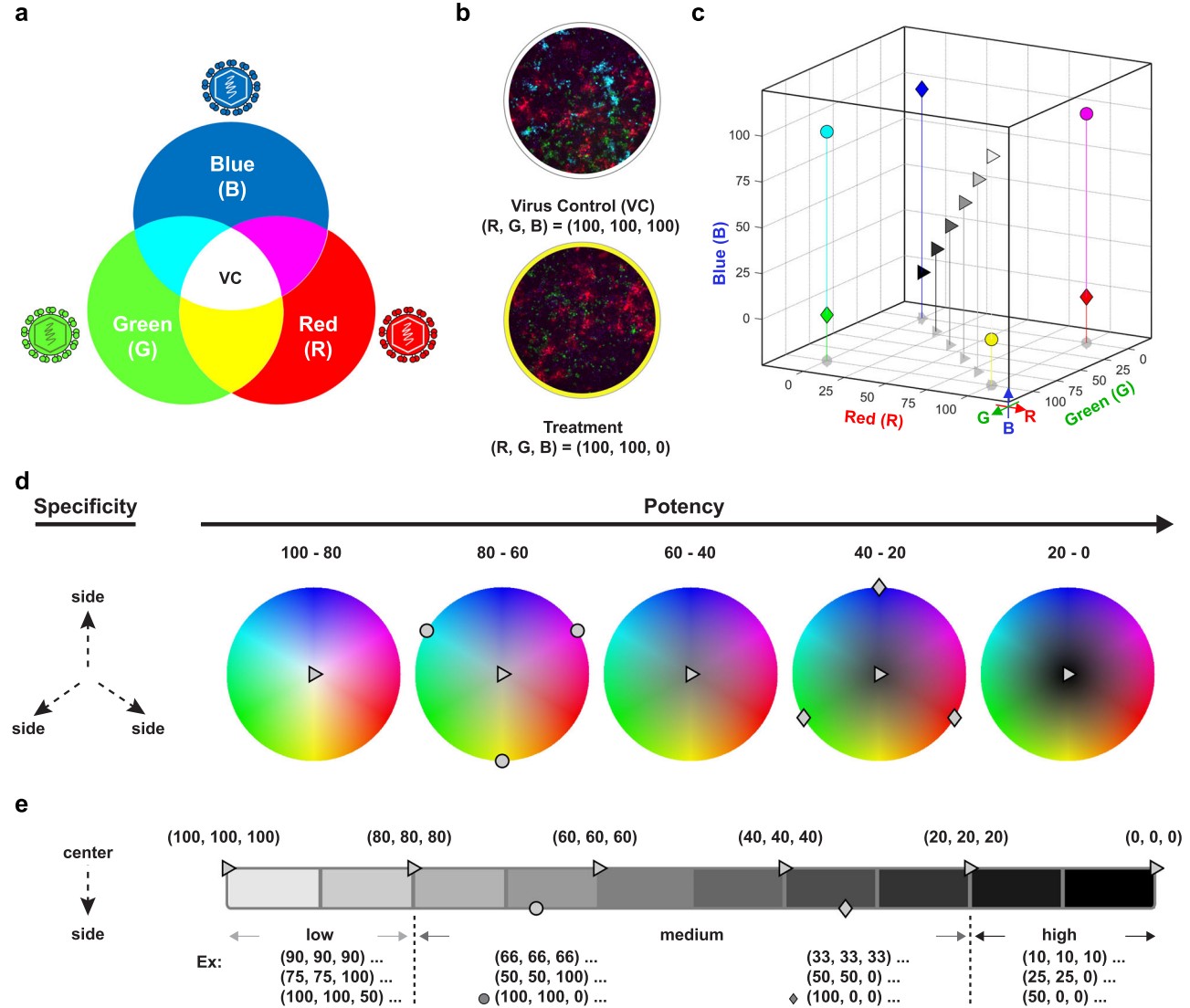

**Fig. 5 | Concept of RGB (Red-Green-Blue) virus color mixing. a** RGB model. Color mixing is used to present the relative infectivity rates of YF17D/mCherry (R), JEV/eGFP (G) and DENV-2/mAzurite (B) in a multiplex-virus assay upon treatment. In case of unrestricted virus growth (virus control, VC), the readout of this color mixing will result in "white", while in case of full inhibition of all three viruses, it will be "black". **b** Showcase of differential compound sensitivities. The relative infectivity for the virus control (upper panel) is computed as white (100, 100, 100). In the theoretical case of a treatment that results in 100% inhibition of only the "blue" virus (lower panel), it will thus yield the complementary color "yellow" (100, 100, 0). **c** 3D plot. When XYZ axes are used to present the three color-coded coordinates (R, G, B), all possible readouts upon treatment can be present in a 3D cube. For example, circles−100% inhibition of only one virus each; diamonds−100% inhibition of each two viruses; black triangle−100% inhibition of all three viruses; white triangle−no inhibition of any of the three viruses. **d, e** Specificity and potency. Each data point in the 3D plot can be deconvoluted on a series of RGB palettes (**d**) to interpret mainly for the specificity, plus a white-black scale (**e**) as representation of potency (**e**) of a particular antiviral molecule under study.

three FPs (mAzurite, eGFP, and mCherry) and three viruses that belong to the same genus. Yet multiplexing beyond three reporters should be feasible, certainly given the rapidly evolving HCI technology. Also, multiplex assays with viruses that represent the tick-borne species within the Orthoflavivirus genus (such as TBEV), belong to different genera, or even families should be feasible to speed-up the discovery of antiviral drugs (and even not per se broader-acting antivirals). Reporter viruses have for example been developed against chikungunya virus/alphaviruses[48], EV-A71/enteroviruses[49], Bunyamwera virus virus/bunyaviruses[39], and SARS-CoV-2/coronaviruses[50,51]. Here, we demonstrate combined infection of cells with viruses representing diverse taxa (Fig. 4), such as parainfluenza virus 1 (PIV1/eGFP, *Paramyxoviridae*, Mononegavirales), chikungunya virus (CHIKV/mCherry, Alphavirus), Bunyamwera virus (BUNV/mCherry, Bunyavirales), and DENV-2/mAzurite (flavivirus). These human RNA viruses cover four distinct viral orders and classes, each with unique replication strategies and markedly different growth kinetics (ex, BUNV > > PIV > DENV-2). The basic requirements for the successful development of such assays are (i) to select cell lines [such as Vero E6 (parent of V-NIR)] that are permissive to a wide array of human viruses, (ii) to carefully balance infection ratios, (iii) to select conditions with no or limited CPE, and finally (iv) to use HCI technology that allows higher spectral resolution than that used in the current study (Supplementary Table 1). Under such premises, the full rainbow spectrum of available FP[52,53] could be exploited for multiplexing.

For the Orthoflavivirus assay, we used reporter viruses based on attenuated vaccine (YF17D; JE SA14-14-2) or lab strains (DENV-2 NGC). Insertion of an FP in the viral genome leads to an additional attenuation of viral replication kinetics and of CPE formation (Supplementary Figs 2 and 3). As demonstrated before[23], such a reduction in cell loss,

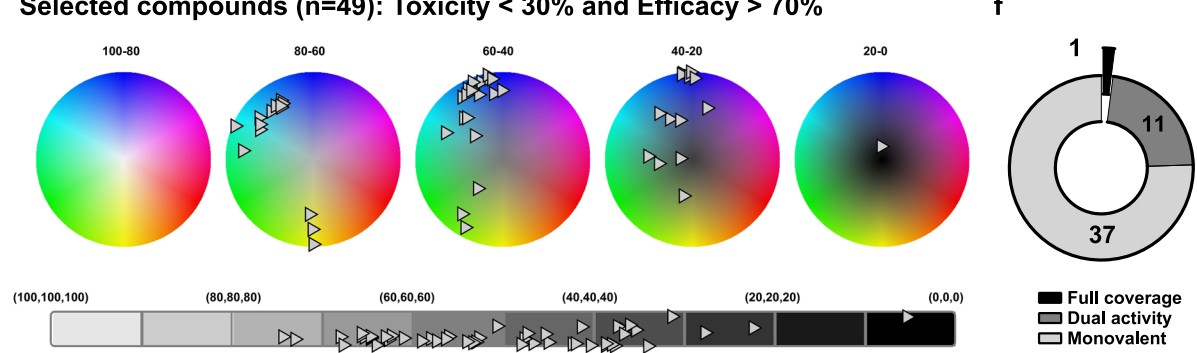

**Fig. 6 | Pilot scale multiplex orthoflavivirus screen.** Performance of reference compounds as shown in (**a**–**c**) as quality control during the screen. Antiviral activity of the five reference compounds, namely 7DMA, BDAA, JNJ-A07, ST-148 and NITD008, was assessed through a series of independent repeat experiments ($n = 5$). **a** The best antiviral activity for each individual reference compound per experiment were plotted on the RGB cube (Supplementary Movie 3) and **b** deconvoluted on series of RGB palettes and white-black scale for visualization of specificity and potency. **c** Dose response curves for reference compounds. The data presented as means ± SD ($n = 5$) from independent experiments. **d** Raw data of all $n = 1256$ compounds subjected to the multiplex screen (plotted as in **b**). **e** Selection of hits ($n = 49$) with sufficient activity (>70% activity against at least one virus). Compounds with unspecific high cytotoxicity (>30% reduction cell count) were excluded. **f** Hit profile with $n = 37$ candidate showing activity to only one virus, $n = 11$ dual-activity and $n = 1$ pan-Orthoflavivirus activity. Source data are provided as a Source Data file, including overlapping data points not readily visible on the graphs.

which caused by CPE, improves assay robustness (Supplementary Fig. 2c and Fig. 2b) and can be considered an extra asset for HCI based scoring. Moreover, use of vaccine strains[54,55] or surrogate viruses for antiviral testing is well established and allows to work under more relaxed biosafety containment conditions (BSL2 instead of BSL3), whereby common drug targets remain unaltered.

In addition to reporter viruses, another key for our multiplexed assay is the use of reporter cells, V-NIR. This cell line was engineered by fusing miRFP703 to histone protein H2B to have an endogenously expressed label for cell nuclei. In contrast to DAPI and other nuclear dyes, signals from miRFP703 can be readily differentiated from the signals of mAzurite, eGFP, and mCherry (Fig. 1d and Supplementary Fig. 1c; Supplementary Table 1). More importantly, the use of an endogenously expressed label that can be monitored without any further staining, facilitates the design and development of homogenous assays, and enables the possibility of continuous live imaging[56]. Likewise, cytotoxicity of the tested compounds will result in a drop in the number of cells expressing near-infrared signals (Fig. 6c, ST-148). Combining reporter viruses and reporter cells, allows to conveniently visualize concomitant infections of multiple viruses in one well and in real time by HCI.

Our multiplexed approach was initially validated using Vero (monkey kidney) cells. Additionally, we also demonstrate its applicability when using human cell lines, such as Huh-7 cells (human hepatoma; Supplementary Fig. 9) and A549 cells (human lung carcinoma; Fig. 4g–i), each having their own benefits and drawbacks. For instance, use of Huh-7 cells may be considered relevant since the human liver is an important target for many viruses, given its role in drug metabolism as well. Likewise, in contrast to many other continuous cell lines, A549 cells remain competent for innate type I IFN signaling, that is critically involved at the virus-host interface. On the other hand, replication efficacy for various viruses in A549-NIR cells was rather low (Fig. 4g, h), requiring markedly higher MOIs for infection which compromise cell viability (Fig. 4i). Yet flexibility regarding choice of various virus-cell combinations as demonstrated herein offers promises for future drug discovery programs. Highly susceptible Vero cells are frequently used in antiviral HTS campaigns to safeguard assay robustness and scalability[40]. As thus identified candidates advance through the various steps of drug development, their further evaluation and optimization requires testing in relevant step-up models, including cells derived from tissues involved in viral pathogenesis[40] and human drug metabolism[57].

Last, to ease and accelerate data analysis for high-throughput library screens, we developed an intuitive kernel based on the RGB color model. By this RGB paradigm, changes in the infection rates of DENV-2/mAzurite, JEV/eGFP, and YF17D/mCherry upon treatment can be converted into a simple color code on a series of RGB palettes plus a WB spectrum (Supplementary Movie 3 and Fig. 5); to readily identify the specificity and the potency of the molecules tested. Each molecule from a multiplexed antiviral screen can hence be readily specified by three Cartesian coordinates (x, y, z) for the ease of internal comparison and ranking. We demonstrate the convenience of this approach by plotting and mining data originating from a pilot, medium-scale multiplex antiviral screen of a library of about 1200 compounds (Fig. 6), delivering ~4% hits (49/1256) with sufficient selectivity (toxicity <30%) and activity against at least one orthoflavivirus (37/1256; ~3%), few with dual activity (11/1256; 0.8%) and one (1/1256; 0.08%) which inhibited all three viruses.

In summary, we showcase an all-optical multiplexed anti-flavivirus assay that is amenable for an imaged-based high-throughput antiviral screening. This study should lay the foundation for the development of other pan-genus, pan-family antiviral screens or even multiplexed antiviral screens using viruses belonging to different genera or families. Such screens should greatly facilitate the development of (broad-spectrum) antiviral drugs.

## Methods

### Ethical statement
Mouse sera used were sourced from immunization experiments conducted at the KU Leuven Rega Institute in accordance with institutional guidelines approved by the Ethical Committee of the Animal Research Center of KU Leuven, Belgium under project numbers P100/2019 (YFV-specific sera) and P140/2016 (DENV2 and JEV-specific sera).

### Cell culture
Baby hamster kidney cells (BHK-21J)[58] and African green monkey kidney cells (Vero E6; ATCC CRL-1586) were maintained in minimum essential medium (MEM, Gibco) with 10% fetal bovine serum (FBS, HyClone) at 37 °C with 5% $CO_2$. Human hepatoma cells (Huh-7; kindly provided by Prof. Ralf Bartenschlager, Heidelberg University, Germany)[59] were maintained in Dulbecco's modified Eagle's medium (DMEM, Gibco) with 10% fetal bovine serum, 20 mM HEPES (Gibco), and 1% nonessential amino acids (NEAA, Gibco); mosquito cells (C6/36; ATCC-CRL-1660) were cultured in Leibovitz's L-15 Medium (Gibco) with 10% FBS, 10 mM HEPES (Gibco), and 100 U/mL penicillin/streptomycin (Gibco) at 28 °C.

Vero E6 and A549 (ATCC CCL-185) cells constitutively expressing near infrared fluorescent protein miRFP-703 (V-NIR and A549-NIR) as translational fusion to histone protein H2B were generated by lentiviral transduction. V-NIR cells were maintained in MEM with 10% FBS and 5 µg/mL blasticidin (Thermo Fisher Scientific), while A549-NIR cells were maintained in DMEM (Gibco) with 10% fetal bovine serum, 100 µg/ml hygromycin, 1 µg/ml puromycin and 10 µg/ml blasticidin. For details on the construction of V-NIR, see Supplementary Methods.

### Reporter viruses
All viruses used were generated recombinantly from infectious cDNA clones. The construction and characterization of mCherry-tagged YF17D (YF17D/mCherry) and eGFP-tagged JEV SA14-14-2 strains (JEV/eGFP) have been reported previously[60–62]. The construction of infectious clone pShuttle-DV2/mCherry expressing the full-length RNA genome of DENV-2 New Guinea C (NGC) strain plus mCherry as N-terminal translational fusion to the DENV-2 polyprotein has been described earlier[23]. Here, a series of derivatives of pShuttle-DV2/mCherry expressing different fluorescent proteins (FP), namely mAzurite (BFP, blue), eGFP (GFP, green), mCitrine (YFP, yellow), and mMaroon (RFP, dark red) were generated in a similar manner, as described in the Supplementary Methods (Supplementary Fig. 1).

All virus stocks were generated in C6/36 cells as described previously[23], except for YF17D/mCherry, which was generated in Vero E6. Virus titers were determined by plaque assays on BHK21J cells[60,61]. Detailed protocols for generation of recombinant DENV-2 variants, virus rescue and culture are provided in the Supplementary Methods.

The generation and use of mCherry-tagged CHIKV[36,37] and BUNV[38,39] have been described before. PIV1 expressing GFP was sourced from ViraTree LLC (Cat. No. P101).

### Serum samples
Virus-specific antisera used in this study have been characterized before[62] and were pools of sera derived from mice vaccinated with either YF17D[61] (Stamaril, Sanofi-Pasteur), JE SA 14-14-2[61], or DENV-2 NGC[23]. All sera were incubated at 56 °C for 30 min to inactivate complement and split into aliquots before storage.

### Serum neutralization tests (SNT)
V-NIR cells ($3 \times 10^4$) were seeded with cultured medium containing 5 µg/mL blasticidin in black 96-well plate (Greiner Bio-One, #655090), and incubated overnight at 37 °C with 5% CO2. On the other day, medium was replaced with MEM medium containing 2% FBS. Titers of the specific neutralizing antibodies in each serum pool were determined by using V-NIR cells and the reporter orthoflaviviruses, namely

DENV-2/mAzurite, JEV/eGFP, and YF17D/mCherry. In brief, individual virus DENV-2/mAzurite, JEV/eGFP, or YF17D/mCherry, or a pre-mixture thereof containing all three viruses were mixed with two-fold serially diluted sera (1:1 volume ratio of virus and serum) and incubated for 75 min at 37 °C. These mixtures were then transferred to V-NIR cells for infection with final multiplicity of infection (MOI) of 0.5 for DENV-2/mAzurite, 0.5 for JEV/eGFP, and 0.2 for YF17D/mCherry. After a 2h-incubation at 37 °C, the virus-serum mixtures were removed and replaced with fresh MEM containing 2% of FBS. At 3 days post infection (dpi), the infection rate was quantified by counting the specific FP-expressing cells (infected cells) versus the total cells by high-content imaging (HCI; Arrayscan XTI, Thermo Fisher Scientific), and relative infection rates were determined by normalizing to virus controls (non-serum treated controls), which were set to 100%. Neutralization curves (log2 values, starting from 1:40 or 1:80) were generated using Graph-Pad Prism as before[23,62]. Serum neutralizing titers resulting in a 50% ($SNT_{50}$) and 90% ($SNT_{90}$) reduction in relative infection rates were derived by nonlinear regression.

### Antiviral assays

V-NIR cells were seeded and cultured as described for the serum neutralization test. In brief, the individual orthoflavivirus, namely DENV-2/mAzurite, JEV/eGFP, or YF17D/mCherry or a pre-mixture thereof was directly added to V-NIR cells for infection at an MOI of 0.5 for DENV-2/mAzurite, 0.5 for JEV/eGFP, and 0.2 for YF17D/mCherry. After a 2h-incubation at 37 °C, the individual virus or the virus mixture was removed and serial dilutions of each antiviral molecule (IFN-α, NITD008, and JNJ-A07) added with MEM containing 2% of FBS. At 3 dpi, individual infection rates were quantified by counting the specific FP-expressing cells (infected cells) versus the total cells by HCI, normalizing infection rates to virus controls (non-treated controls) set to 100%. Inhibition curves (log10 values) were generated using GraphPad Prism 8, and the respective concentrations resulting in 50% ($EC_{50}$) and 10% relative infection rate ($EC_{90}$) were determined by a nonlinear regression method.

### Image acquisition

Image acquisition was performed using a 7 color-engine widefield high-content imager (Arrayscan XTI, Thermo Fisher Scientific). Pixel resolution was set to 1104 × 1104 with a 2 × 2 binning to minimize data storage while ensuring image quality. A 10× objective capturing 9 fields/well was chosen to ensure sufficient image feature resolution while retaining scanning speed and optimizing data storage. In particular, the restriction in the number of fields captured enough cells for statistical analysis, while delivering relatively fast post image acquisition processing time. Six imaging channels were used for the primary multicolor assay. Each channel contains specific settings for excitation wavelengths, dichroic mirror placement, emission filters and exposure times (Supplementary Table 1). Combinations of custom dichroic mirror and emission filters were systemically tested and optimized to achieve minimal bleed-through signal between channels. Exposure times were determined by setting the brightest pixel intensity value of the image on 50% of the 14-bit dynamic range from the CCD (charge-coupled device) camera. Exposure times were checked throughout experiments and readjusted when necessary to compensate for the increasing fluorescence intensities resulting from viral replication and intracellular accumulation of FPs over time.

### Image analysis on final four colors

Image analysis and data processing was performed by using the HCS Studio Cellomics software (Thermo Fisher Scientific, Version 6.6.2). The Spot-Detector Bioapplication was chosen as a powerful multi-mode pixel intensity measurement tool for quantitative analysis. Images on all four selected custom channels (V-NIR, Azurite, GFP and Cherry as shown in Supplementary Table 1) were first processed with a

background reduction using the "*3D_SurfaceFitting*"-method which removes unspecific noise while retaining dim edges of the fluorescent signal. Background removal settings were determined by assessing image intensity histograms and set to remove the background peak. The following step was to determine and identify ValidObjects (=nuclei of the cells) in the near infrared channel (NIR signal in the nucleus due to fusion of histone protein H2B). A fixed threshold was used to identify objects and simultaneously a soft smoothing and segmentation algorithm was applied to acquire accurate single nuclei as an object in the images. After object identification, object size was set as validation parameter to remove small unspecific objects and large unsegmented or false objects.

The three fluorescent reporter viruses were detected individually in channels 2 (green), 3 (red) and 4 (blue). To that end, Region of Interest (ROI) was selected by using a *Circ mask*-type, widened by several pixels for accurate detection of the virus FP signals residing in the cell nuclei and the cytoplasm. The fluorescent identification or *'spot detection'* started with a fixed threshold which was determined by the lowest non-background signal for each channel. If necessary, the parameter was adjusted during inter-plate experiments to exclude false signal or to include weak signal caused by biological variation (different expression levels of respective FPs) of the fluorescent viruses.

### RGB model algorithm

The RGB model was executed using MATLAB (Version 2017b, The MathWorks, Inc., Natick, Massachusetts) with an in-house developed software (see also code availability section for more details). The model plots the data in view of a 3D cube, plates and a white to black (WB) linear scale. In a cube, all the relative infectivity rates regarding to a compound treatment are denoted as [Mx, My, Mz], where Mx = Redness, My = Greenness, and Mz = Blueness. The data could then be denoted as [M*plate*, M*angle*, M*radius*] by the process below, where M*plate* denotes the index of the plate, M*angle* is the angle of polar coordinate, and M*radius* is the radius of polar coordinate. First, we divided the cube into 5 sessions, says 0 to <20%, 20% to <40%, 40% to <60%, 60% to <80% and 80%-100%. Each session is parallel to the other and perpendicular to the Black-To-White axis (the vector of [0, 0, 0] to [100, 100, 100]). In practice, all data is projected to the Black-To-White axis, and it could be denoted as M*proj*:

$$Mproj = [Mx, My, Mz] \cdot [100, 100, 100] / |[100, 100, 100]| \quad (1)$$

M*plate* is the portion of the length of M*proj*, thus it could be formulated as:

$$Mplate = |Mproj| / |[100,100,100]| * 5 \quad (2)$$

Then, we converted the data from cartesian coordinate [Mx, My, Mz] to polar coordinate [M*angle*, M*radius*] by using the function "rgb2hsv" in Matlab. The M*angle* is the HUE of the color of data, and M*radius* is the product of SATURATION and VALUE of the color of data. It could be written:

$$Mangle = HUE \quad (3)$$

$$Mradius = SATURATION * VALUE \quad (4)$$

We use the following process to plot the data on the WB intensity scale. In the scale, the data could be denoted as [M*length*, M*sat*]. M*length* is the length of M*proj*, and M*sat* is the product of

SATURATION and VALUE of the color regarding to the data. It could be written:

$$Mlength = |Mproj| \tag{5}$$

$$Msat = SATURATION * VALUE \tag{6}$$

## Statistics and reproducibility

No statistical method was used to predetermine sample size. No data were excluded from the analyses. GraphPad Prism 10.0.0 or earlier versions was used for statistical analysis. Unless explicitly specified in the figure legends or outlined below, all experiments were repeated three times, and data are represented as means ± SD from 3 independent biological repeats. For kinetics of individual reporter flavivirus infections (Fig. 1e), data are represented as means ± SD from 4 independent biological repeats. For assessing assay robustness (Fig. 2b), Z′ values were determined from six replicates ($n = 6$) at both 3 dpi and 4 dpi. In the case of multiplex infections presented in Fig. 4, two independent experiments were performed for each combination, except for the combination of DENV-2/mAzurite, PIV1/eGFP, and CHIKV/mCherry, which was performed only once. In the pilot-scale library screen (Fig. 6a–c), five reference plates were used. All representative micrographs presented in this study were from experiments which repeated at least three times and yield similar results. P values were calculated by using one-way ANOVA with Tukey's post hoc test. $P < 0.05$ were considered statistically significant (*$=p < 0.05$; **$=p < 0.01$; ***$=p < 0.001$; ****$=p < 0.0001$). Assay quality (Fig. 2) was assessed as Z′ value for each virus (DENV-2/mAzurite, JEV/eGFP, or YF17D/mCherry) as follows; considering the dynamic range of signals between virus control (VC) and cell control (CC), and the respective variation of data within each group in each custom detection channel (i.e., Azurite, GFP, or Cherry; Supplementary Table 1):

$$Z' = 1 - \frac{(3\sigma_{VC} + 3\sigma_{CC})}{|\mu_{VC} - \mu_{CC}|} \tag{7}$$

where σ represent the standard deviation of the counts, and μ represent the mean of the counts. Z′ > 0.5 is considered amenable for HTS[63].

Results obtained by single-virus and multiple-virus assays were compared by using (i) Pearson's correlations with linear regression analysis, and (ii) the Bland–Altman plots presenting the differences of two paired measurements, with each against the means of these two measurements.

Experiments in this study were not randomized as a fixed layout was used on each scanning plate with cells assigned to either the predetermined experimental or control groups. No specific group allocations were involved in this study, the concept of researcher blinding was not applicable. Notably, the investigators did not receive information regarding the compound library used in the HTS and the identity of the respective small molecules remained undisclosed throughout the study.

### Reporting summary

Further information on research design is available in the Nature Portfolio Reporting Summary linked to this article.

## Data availability

All relevant data of this study are available in the paper and its supplementary materials. Source data are provided with this paper. The raw and full-resolution images are available upon request due to large data size constraints. Any additional requests for information can be directed to the first author and the corresponding author. Such requests would be fulfilled within a week. Source data are provided with this paper.

## Code availability

The code written for automated data analysis and visualization is deposited at GitHub and can be accessed through GitHub (https://github.com/YunAnGitHub/RGB_Virus_Model). This package has also been deposited at Zenodo: https://zenodo.org/records/10113251[64].

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

## Acknowledgements

We thank Ruben Pholien, Joost Schepers, and Thibault Francken for their excellent technical support on HCI screening; Dr. Ji Ma for collecting serum samples from DENV-2, JEV, or YF17D-infected IFNAR mice; Catherina Coun and Nathalie Thys (TPVC) for construction of JEV/eGFP virus, and V-NIR and A549-NIR cells; and Katrien Geerts for support with virus replication kinetics. We are grateful to Prof. Andres Merits (University of Tartu, Estonia) for sharing the infectious clone of CHIKV/mCherry, and Dr. Rana Abdelnabi and Prof. Leen Delang (KU Leuven Rega) for rescuing respective virus stocks. We are grateful to Prof. Xiaohong Shi (University of St. Andrews, Scotland, UK) for sharing BUNV/mCherry virus; and Tina van Buyten and Dr. Dirk Jochmans (KU Leuven Rega) for sharing A549-NIR cells and PIV1/eGFP virus stocks. PIV1/eGFP created by Dr. Emmalene J. Bartlett, Dr. Peter L. Collins and Dr. Brian R. Murphy (National Institutes of Health, US) was kindly provided via Vira-Tree. Huh-7 cells were kindly provided by Prof. Ralf Bartenschlager, Heidelberg University, Germany. This project received funding from the European Union's Horizon 2020 Research and Innovation Program under grant agreement no. 733176 (RABYD-VAX to K.D. and J.N.) and from the European Union's EU4Health program under grant agreement 101102733 (DURABLE to J.N.). Funded by the European Union. Views and opinions expressed are, however, those of the authors only and do not necessarily reflect those of the European Union. Neither the European Union nor the granting authority can be held responsible for them. Further funding was provided by the Research Foundation Flanders (FWO) under the Excellence of Science (EOS) program (no. 30981113, VirEOS to K.D. and J.N.), and the FWO Hercules Foundation [grant number ZW13-02, Caps-It infrastructure]. Part of the research was funded by the Flanders Agency Innovation & Entrepreneurship (VLAIO grant HBC.2021.1131). KU Leuven provided funding support to L.H.L. (PhD scholarship, DBOF/14/062), S.T.H. (C24/18/080), J.R.P. (Starting Grant, STG/21/028), and K.D. (C3/19/057 Lab of Excellence).

## Author contributions

Conceptualization: L.H.L. and K.D.; Data curation: L.H.L. and W.C.; Analysis: L.H.L., W.C., and Y.A.H.; Funding acquisition: K.D., J.N., and P.L.; Investigation: L.H.L., W.C., and Y.A.H.; Methodology: L.H.L., W.C., Y.A.H., S.J.F.K., S.T.H., T.V., M.R. and H.J.T.; Resources: P.L. and J.N. (Caps-IT), G.V., D.B., O.G. (compound library), and J.R.P. (reagents); Software: Y.A.H. and L.H.L.; Supervision: K.D.; Validation: L.H.L. and K.D.; Visualization L.H.L.; Writing L.H.L. and K.D.; Reviewing and editing by all authors.

## Competing interests

G.V., D.B. and O.G. are full employees of Janssen Pharmaceutica NV (a Johnson & Johnson Company) and may hold stocks or stock options of the company. All other authors declare no competing interests.
