## [Peer Review File · Nature Communications]

Reviewers' comments:

Reviewer #1 (Remarks to the Author)

In this manuscript, Li et al. developed a cell-based multiplex/multicolor antiviral HCS/HTS assay, amenable to high-throughput, and validated the assay using virus-specific antisera and known small-molecule inhibitors. A pipeline/kernel was developed suitable to deconvolute the resulting multidimensional quantitative data.

Overall, the experiments are very well-designed, the data are clear, and the manuscript is well-written.

This elegant and original work is of importance and will accelerate the discovery of novel viral interventions by screening multiple distinctly related viruses simultaneously; however, there are a few minor shortcomings that the authors should address.

The comments given below may help revise this article.

Discrepancies

- Figs. 4E/5B: The greyscale (WB) representation is adding another layer of complexity without providing additional information. Figs. 4D/5A are extremely informative and an elegant representation of complex/multiplex data. The information on which virus is inhibited is lost in greyscale (Fig. 4D).
- Line 204: WB scale for overall potency, from “low” to “high” (Fig. 4E; left to right) on the “x-axis.” What about the “y-axis” of Figs. 4E/5B? Please add further explanations in the text and Fig. legends or consider moving the WB representation into the supplement.

Minor discrepancies

- Line 63: change toxicity to cytotoxicity
- Line 75: change Mutliplexed to Multiplexed
- Line 111: DV2/mAzurite had the slowest replication kinetics. Is this due insertion of the FP?
- Line 118: Please speculate why no co-infection is detectable. Is there no viral spread?
- Consider moving Fig. 1A into supplement
- Line 551: change High Content Imaging (HCI) to HCI
- Line 553: change Represenative to Representative
- Fig. 1E: Please explain why MOI of 0.1 results in higher infection rates as compared to 0.2 and 0.5.
- Fig. 3A: Please explain the increase of YFV infection in the presence of anti-JEV antibody

- Table 2: Please change IFN to IFN α in the table
- Fig. 3D: IFN α treatment of JEV and YFV, no sigmoidal curves. EC50 values are extrapolated?
- Fig. 4C: please label the axes
- Suppl. Fig. 3: change μ M to μ M and nM to nM

Reviewer #2 (Remarks to the Author)

The manuscript entitled “Multiplexed multicolor antiviral assay amenable for high throughput research” by Li-Hsin Li et al. developed a high-content imaging cell-based multiplex/multicolor antiviral assay. The manuscript provides sufficient advances in approach towards antiviral drugs that seem to be rather active against an entire genus or even family of viruses. Though the study is well designed with sufficient proof in this direction, it is still unclear how this approach will lead to the identification of novel and potent anti-viral drugs. Moreover, there are several efforts including <https://www.frontiersin.org/articles/10.3389/fviro.2022.854363/full>, which utilized single fluorescent proteins (FP) based screening of anti-viral drugs. Overall, this study very elegantly provides pieces of evidence for a multiplexed multicolor antiviral assay that may be further utilized for other virus combinations.

Still, there are a few concerns that will be nice to answer, before its publication

- 1) This study utilized a genus of the flaviviruses using three distantly related flaviviruses that include the dengue, Japanese encephalitis, and yellow fever virus each tagged with spectrally distinct fluorescent proteins (FP) [respectively mAzurite, eGFP, mCherry], As these viruses show distinct time kinetics and levels of infections, it is not very convinced for the time of exposure of compounds / anti-viral that were continuously present, wash and replenish every 24 h etc. As active state and threshold levels of compounds are essential for exhibiting potency. Though it is encouraged that authors compared the assay with individual tagged FPs.
- 2) Authors claimed to develop a kernel based on RGB (Red-Green-Blue) paradigm for antiviral activities, though data and presentation do not look very promising towards its application, as it is not clear and intriguing.
- 3) Overall, the study is elegant and with sound methodology, and also validates data using antisera containing virus-specific neutralizing antibodies and pan-flavivirus inhibitors, interferon alpha (IFN- α), and NITD008.
- 4) The only challenge seems to titration of viral titer to be used for such multiplexing for other viruses.

Reviewer #3 (Remarks to the Author)

Li and colleagues present here a multiplex multicolor approach that will allow the screening of small molecules libraries in the context of co-infection with several viruses. The screen will be performed in modified simian cells engineered by the team, allowing quantitation of cellular toxicity of the tested compounds. They used 3 fluorescent reporter flaviviruses and previously known flaviviral inhibitors to set up their assay and demonstrate feasibility.

Overall, the results are convincing and the experiments are well executed. The paper is well written.

Major issues:

I feel that the assay should be used in the context of a genuine screen to represent a significant advance. A manuscript describing solely the methodology seems preliminary to me for a publication in Nature Communications.

Demonstrating the feasibility of the assay using viruses belonging to different families (with different kinetic of replication) would be a nice addition.

The use of simian cells is questionable. While these cells have the advantage of supporting replication of a wide array of viruses, the toxicity of a given compounds is unlikely to be the same in simian cells than in human cells.

Minor issues:

- TBEV is not a mosquito-borne virus (lines 45-46)
- The authors claim that the modified DENV carrying fluorescent proteins replicate similarly. However, it seems to me that they have very different fitness (the size of the plaques shown in sup figure 1 are very different). Such difference in fitness may influence the potency of the identified compounds.
- Please justify the use of YFV and JEV vaccine strains.
- What about fitness of FP-YFV and FP-JEV as compared to unmodified counterpart?
- Line 32: please add references for HIV/HCV antiviral drugs (reviews will do)
- The 3 selected Flaviruses are related mosquito-borne viruses. Results may not be extrapolated to more distant flaviviruses, such as TBEV for instance.
- Fig 1F. Please indicate the MOI and the time at which the experiment was done.

Typos:

Line 161: NS4B instead of N4B

Reviewers' comments:

Reviewer #1 (Remarks to the Author)

In this manuscript, Li et al. developed a cell-based multiplex/multicolor antiviral HCS/HTS assay, amenable to high-throughput, and validated the assay using virus-specific antisera and known small-molecule inhibitors. A pipeline/kernel was developed suitable to deconvolute the resulting multidimensional quantitative data.

Overall, the experiments are very well-designed, the data are clear, and the manuscript is well-written.

This elegant and original work is of importance and will accelerate the discovery of novel viral interventions by screening multiple distinctly related viruses simultaneously; however, there are a few minor shortcomings that the authors should address.

The comments given below may help revise this article.

Many thanks for the enthusiastic appreciation of our work. As explained in detail in the subsequent sections, we refined our description on how to project and analyze the multidimensional data. By this, we hope to provide a more intuitive insight into the data mining process.

Discrepancies

- Figs. 4E/5B: The greyscale (WB) representation is adding another layer of complexity without providing additional information. Figs. 4D/5A are extremely informative and an elegant representation of complex/multiplex data. The information on which virus is inhibited is lost in greyscale (Fig. 4D).

The added value of the greyscale (WB) representation may have not been fully clear. Therefore, we have created a new Supplementary Video 3. This video aims to provide a more comprehensive illustration and visualization of our 3D model, with a specific focus on explaining the origins of both the RGB palettes and the WB projections.

While RGB serves as proxy for representing “virus-specificity”, WB does so for assessing “potency”. In simple terms, consider that a hit compound can be effectively powerful in switching off one particular fluorescence signal, indicating mono-specific antiviral activity. However, this might not significantly reduce the overall brightness of all fluorescence signals because it might lead to a failure in inhibiting the other two viruses (colors). Consequently, in such case, the overall brightness is dimmed by a maximally of 33%, resulting in a position on the left bottom side of the WB panel. For a future antiviral, this would translate into a failure to protect against all pathogens by said inhibitor, i.e., limited breadth of activity (hence score “W” = “all/most viruses ON” than “B” = “all/most viruses OFF”).

Also, we add a full new data set originating from a pilot-scale screen of about 1000 drug-like small molecules in our multiplex assay (**new Figure 6**). Firstly, this exercise showcases the feasibility and potency of the multiplex approach. Furthermore and at least for us most convincing, plotting the compiled >1000 data points into only two panels (color and grey scale) may serve as an illustrative example and demonstrate how both representations complement each other: (i) hits with high specificity yet limited breadth of inhibitory activity (color), (ii) hits

with promising pan-flavivirus antiviral activity (greyscale), and (iii) elimination of toxic compounds (cell count).

Please also see our narrative describing these results in lines 269-291.

- Line 204: WB scale for overall potency, from “low” to “high” (Fig. 4E; left to right) on the “x-axis.” What about the “y-axis” of Figs. 4E/5B? Please add further explanations in the text and Fig. legends or consider moving the WB representation into the supplement.

As shown on revised Fig. 5d, e (old Fig. 4) and Supplementary Fig. 10 (old Fig. 5), we added the relevant information with arrows in dashed line. For the RGB palettes, the arrows show the level of specificity from center to side; and for the WB scale, the arrow from the top to the bottom (equal to center to side on the RGB palettes) represents the level of the specificity. The legend has been expanded accordingly. Please also see our response to the previous comment.

Please also see our narrative describing these results in lines 252-256.

Minor discrepancies

- Line 63: change toxicity to cytotoxicity

Done as suggested (line 75).

- Line 75: change Mutlplexed to Multiplexed

Done as suggested.

- Line 111: DV2/mAzurite had the slowest replication kinetics. Is this due insertion of the FP?

As demonstrated previously by us (REF #23), there are no significant differences on virus yields between DENV-2/WT and DENV-2/mCherry until 4 dpi. In this manuscript, we investigated the virus replication competence of more DENV-FP variants by using two different MOI, 0.1 and 0.01. The results show a consistent attenuation of all the DENV-FP variants based on the phenotype of their smaller plaque sizes compared to the DENV-2/WT (Supplementary Fig. S1b). However, the replication kinetics of all DENV-FP variants remain quite similar, with the only exception might be the mMaroon variant, which shows reduced progeny yield (Supplementary Fig. S2). Based on these results, we conclude that insertion of any foreign sequence leads to the attenuation of DENV-2/WT, while overall growth characteristics remain almost unaltered. Additionally, a similar effect was observed when comparing YF17D/mCherry and JEV/eGFP with their respective parental viruses (new Supplementary Fig. S3). Taken together, the trait that DENV2/mAzurite had the slowest replication kinetics among the DENV-2/mAzurite, JEV/eGFP and YF17D/mCherry is likely inherent to the virus strain itself and not due to the insertion of the FP.

- Line 118: Please speculate why no co-infection is detectable. Is there no viral spread?

Flaviviruses (as many other RNA and DNA viruses) show strong superinfection exclusion disabling secondary infections during virus spread in mixed infections. We add this notion including a series of new references (**line 142-146**). Nevertheless, there is clear evidence for virus spread; visible as an increase of number of infected cells over time (**see e.g., Fig. 1e, Supplementary Fig. 2d, Supplementary Fig. 3c,d**).

- Consider moving Fig. 1A into supplement

We prefer to keep this panel as it demonstrates the feasibility to use and combine FP deliberately. Please see our previous response (**line 96-106**). The aim is to show the flexibility and (almost) free choice of FP for multiplexing.

- Line 551: change High Content Imaging (HCI) to HCI

Done as suggested (**line 759**).

- Line 553: change Represenative to Representative.

Done as suggested (**line 761**).

- Fig. 1E: Please explain why MOI of 0.1 results in higher infection rates as compared to 0.2 and 0.5.

This notion is highly appreciated. An error occurred in labeling the symbols, which are now corrected.

- Fig. 3A: Please explain the increase of YFV infection in the presence of anti-JEV antibody

The viruses seem to interfere at population level in multiplex infection. Similarly, we see that the relative infection rates of JEV/eGFP increased when YF17D/mCherry is inhibited (**Fig. 3a, anti-YFV Ab**). It is also observed occasionally when we perform the antiviral assays (**Fig. 3d, IFN- α and new Fig. 6, BDAA and JNJ-A07**) that if not all the viruses are inhibited, those viruses that are less sensitive to the inhibitor gain a relative advantage in the multiplex assay (higher relative replication fitness). In terms of population ecology, “the capacity of the habitat” for one virus may get expanded when the others are eliminated as “sympatric competitors”. Obviously, some inhibitors pivot a dynamic equilibrium that is established in untreated virus controls.

Most importantly, the thus obtained SNT_{50} values remain in a reasonable range comparable to those obtained in the respective single-plex assay (**Table 1 and Fig. 3c**). Additionally, the efficacy and potency of the inhibitor remain unchanged as the position of JNJ-A07 on the RGB model remains the same with or without this phenomenon (**Fig. 6b and new Supplementary Fig. 9**). This information is also added in the manuscript (**line 176-178**).

- Table 2: Please change IFN to IFN α in the table

Done as suggested.

- Fig. 3D: IFN α treatment of JEV and YFV, no sigmoidal curves. EC50 values are extrapolated?

In this manuscript, IFN- α was used as a host-targeting reference compound to validate our multiplex screening assay. The results show distinct sensitivities of DENV-2/mAzurite, JEV/eGFP, and YF17D/mCherry to IFN- α treatment during viral infections. For instance, JEV/eGFP and YF17D/mCherry show greater sensitivity to the treatment, with a minimal tested concentration of about 1 IU/mL already resulting in some reduction in viral replication compared to the untreated controls (set as 100%). Consequently, the determination of EC50 values is extrapolated. The information is now added as footnote to **Table 2 (footnote b)**.

- Fig. 4C: please label the axes

Done as suggested (**now new Fig. 5d,e**).

- Suppl. Fig. 3: change μ m to μ M and nm to nM.

Done as suggested.

Reviewer #2 (Remarks to the Author)

The manuscript entitled “Multiplexed multicolor antiviral assay amenable for high throughput research” by Li-Hsin Li et al. developed a high-content imaging cell-based multiplex/multicolor antiviral assay. The manuscript provides sufficient advances in approach towards antiviral drugs that seem to be rather active against an entire genus or even family of viruses. Though the study is well designed with sufficient proof in this direction, it is still unclear how this approach will lead to the identification of novel and potent anti-viral drugs. Moreover, there are several efforts including <https://www.frontiersin.org/articles/10.3389/fviro.2022.854363/full>, which utilized single fluorescent proteins (FP) based screening of anti-viral drugs. Overall, this study very elegantly provides pieces of evidence for a multiplexed multicolor antiviral assay that may be further utilized for other virus combinations.

Many thanks for the encouraging comments. Before answering the specific questions, we would like to briefly explain why we believe that our novel high-content imaging based multiplex reporter virus approach could be a major advancement in the field aiming at the identification of novel and potent anti-viral drugs with a broad(er) spectrum of activity. Using the forementioned study as an example of a very typical “classical” screening campaign. Therein the authors used virus-induced CPE (reduction in cells counts due to virus infection) as a phenotype for screening of 400 compounds, scoring “loss in number of cells” as proxy for failure to inhibit viral replication. Likewise, viral replication was only detected under the promise that the thus tested virus caused CPE (which is not the case for many human pathogenic viruses, in particular clinical isolates), and counting of cells required multiple extra liquid handling steps (fixation and staining of the cells). The evidence that the hits interfere directly with viral replication is only provided later by a reduction in virus yields (plaque assay and RT-qPCR) and by detection of virus antigen (immunofluorescent assay). Unwanted cytotoxicity can only be scored independently from any possible antiviral activity (toxic compounds mimic a strong CPE).

In our multiplex assay, all these pieces of information are easily gathered at once. Firstly, use of reporter cells and reporter viruses facilitates a homogenous assay setup and a continuous optical readout without extra staining steps. As virus-induced CPE is no longer required for read-out, cell-counts can be used to score cytotoxicity. Secondly, our multiplex assay can test a given compound library simultaneously against several different viruses at once. This facilitates discovery of inhibitors with antiviral activity against more than one particular virus. These benefits are also described in the context **(line 63-78 and line 339-349)**.

Besides, for conceptual proof and full support of our approach, we now add a new data set originating from a genuine screen of a library of about 1000 drug-like small molecules in our multiplex assay **(new Fig. 6)**; to showcases the feasibility and potential of our multiplex approach as a novel screening tool for the discovery of hit compounds with broad(er) spectrum antiviral activity.

Still, there are a few concerns that will be nice to answer, before its publication.

1) This study utilized a genus of the flaviviruses using three distantly related flaviviruses that include the dengue, Japanese encephalitis, and yellow fever virus each tagged with spectrally distinct fluorescent proteins (FP) [respectively mAzurite, eGFP, mCherry], As these viruses

show distinct time kinetics and levels of infections, it is not very convinced for the time of exposure of compounds / anti-viral that were continuously present, wash and replenish every 24 h etc. As active state and threshold levels of compounds are essential for exhibiting potency. Though it is encouraged that authors compared the assay with individual tagged FPs.

Viruses with different growth kinetics. Distinct growth kinetics of the individual viruses can be compensated by using different MOIs. Viruses are thus inoculated at the same time (synchronized infections), with a MOI of 0.5 for DENV-2/mAzurite, 0.5 for JEV/eGFP, and 0.2 for the YF17D/mCherry. By this means viruses with very different growth kinetics can successfully be multiplexed. This is now demonstrated for two additional combinations: (1) DENV-2, parainfluenza virus 1 (PIV1), and chikungunya virus, or (2) DENV-2, PIV1, and Bunyamwera virus (BUNV) (**new Fig. 4; narratives in lines 203-215 and 318-324; see also our response to comment 4 below**).

There is no doubt that the active state and threshold levels of the compounds are essential for exhibiting potency. However, a high-throughput screening (HTS) is meant to find a few “needles in a haystack” in a most efficient way and at scale. Thus, we initially focus on specificity and efficacy. Other characteristics, such as stability (half-life), the pharmacokinetics (PK) and Pharmacodynamics (PD), and the mechanism of action, will be validated further in the confirmation steps, following the entire drug development pipeline (**discussed in lines 362-365 and new Ref #40 and #57**).

Compound replenishment. There is little reason to assume that a shortened half-life or replenishment of compounds would have affected the three viruses differentially. Most importantly, the requirement to replenish the test compounds frequently would lower the feasibility, cost effectiveness and sustainability of any HTS assay. Replenishment would multiply (*i*) the need of compound and (*ii*) steps of pipetting/liquid handling. To remain viable, a HTS (aim is to screen $>10^5$ small molecules) should be kept as homogenous as possible limiting the number of manipulations. Secondly, not replenishing compounds may automatically select for hit compounds with sufficient chemical stability (elimination of fast decaying chemical structures).

The assays with individual tagged FPs have been compared (**Fig. 3 and Supplementary Fig. 4**) and the respective values reported for direct head-to-head comparison in **Table 1 (sera)** and **Table 2 (reference compounds)**. In general, values for single and multiplex assessment correlated well as shown in **Fig. 3f (Pearson and Bland-Altman)**.

2) Authors claimed to develop a kernel based on RGB (Red-Green-Blue) paradigm for antiviral activities, though data and presentation do not look very promising towards its application, as it is not clear and intriguing.

We add an example of a pilot-scale screen to support the use of our model and kernel. By using this model, the overall effectiveness of the entire set of 1256 tested compounds can be easily recognized from the left to the right (**new Fig. 6d**). Moreover, by applying filters for toxicity (TOX < 30%) and efficacy (> 70%), potent compounds can be easily selected and visualized on the RGB palettes and the WB scale, showing the specificity (mainly on RGB palettes) and potency (mainly on WB scale) at once (**new Fig. 6e**).

We are convinced that this scoring approach greatly facilitates the selection of potential hits in three key categories: (*i*) hits with promising pan-Orthoflavivirus activity, (*ii*) hits that retain

promising due to their high specificity even if their inhibitory activity does not cover all tested viruses (based on color changes), and *(iii)* a means to eliminate toxic compounds (determined by cell count). We are confident that this example improves the understandability of the concept and makes it more intriguing. In our hands, the kernel worked like a "push button" application for convenient data extraction.

Please also see our narratives describing **(lines 282-291)** and discussing these results **(lines 372-377)**.

3) Overall, the study is elegant and with sound methodology, and also validates data using antisera containing virus-specific neutralizing antibodies and pan-flavivirus inhibitors, interferon alpha (IFN- α), and NITD008.

Many thanks for appreciating the quality of our work.

4) The only challenge seems to titration of viral titer to be used for such multiplexing for other viruses.

We agree and we included the respective data shown in Fig.1e,g and Fig. 2.

For example, we established two additional experimental setups: (1) a combination of PIV1, chikungunya virus and DENV-2, and (2) a combination of PIV1, BUNV and DENV-2 (as shown in **new Fig. 4**). Thereby we demonstrate the feasibility of multiplexing viruses from different families (such as paramyxoviruses/Mononegavirales, alphaviruses, Bunyvirales, and orthoflaviviruses), **covering viruses with different replication strategies and kinetics (lines 203-215 and 319-324)**.

Reviewer #3 (Remarks to the Author)

Li and colleagues present here a multiplex multicolor approach that will allow the screening of small molecules libraries in the context of co-infection with several viruses. The screen will be performed in modified simian cells engineered by the team, allowing quantitation of cellular toxicity of the tested compounds. They used 3 fluorescent reporter flaviviruses and previously known flaviviral inhibitors to set up their assay and demonstrate feasibility.

Overall, the results are convincing, and the experiments are well executed. The paper is well written.

Major issues:

I feel that the assay should be used in the context of a genuine screen to represent a significant advance. A manuscript describing solely the methodology seems preliminary to me for a publication in Nature Communications.

To address the concern, we added a full new data set (**new Fig. 6**). This dataset originates from a pilot-scale screen of around 1000 drug-like compounds in our multiplex assay. Additionally, we introduced three new reference compounds with known mechanism and spectrum of activity: 7DMA, a general (+) RNA virus inhibitor; ST-148, a DENV inhibitor targeting the capsid protein; and BDAA, a YFV inhibitor targeting NS4B.

As shown in **new Fig. 6**, all reference compounds showed the expected responses. The respective reference plates were included multiple times during the screening campaign; serving as experimental controls and benchmark, and as quality controls for assay robustness. After ruling out those candidates that caused a reduction of primary cell counts by more than 30% (due to toxicity) and those with efficacy below 70% (**Fig. 6d-e**), the library screen identified 49 hits, including 11 with dual antiviral activity and one with potentially potent pan-Orthoflavivirus antiviral activity.

Please also refer to the relevant narrative that has been included in the expanded Results section (**lines 270-291**) and Discussion section (**lines 372-377**). Corresponding adjustments have been made to the Abstract (**lines 35-36**) and Introduction (**lines 87-89**).

Demonstrating the feasibility of the assay using viruses belonging to different families (with different kinetic of replication) would be a nice addition.

As requested, we establish another two multiplex infections: (1) a combination of human parainfluenza virus 1 (tagged with eGFP; *Paramyxoviridae*), chikungunya virus (tagged with mCherry; *Alphaviridae*), and DENV-2, and (2) a combination of parainfluenza virus 1, Bunyamwera virus (*Bunyaviridae*) and DENV-2 (**new Fig. 4**). This demonstrates that establishing multiplex virus assay for unrelated viruses with distinct replication strategies and kinetics should be feasible by carefully titrating assay conditions.

Please also see our narrative describing these results (**lines 203-215**) and their discussion (**lines 318-324**).

The use of simian cells is questionable. While these cells have the advantage of supporting replication of a wide array of viruses, the toxicity of a given compounds is unlikely to be the same in simian cells than in human cells.

As mentioned by the reviewer, Vero cells are frequently used for first-line antiviral screening leveraging on their high susceptibility to viral infection, which facilitates HTS, making them robust, reliable, and scalable (**REF #40**). We fully agree that validation in cell lines with higher relevance regarding human host and viral tissue tropism as well as drug metabolism are needed. We now demonstrate that in principle other cells, such as A549 (human lung carcinoma) cells (**new Fig. 4g**) and Huh-7 (human hepatoma) cells (**Supplementary Fig. 9**) could also be used. Either system has their own benefits and drawbacks. E.g., Vero cells are known to express plenty of membrane pumps possibly reducing intracellular drug exposure; Huh-7 show a high genomic instability posing a challenge for assay consistency during large HTS campaigns, and A549 cells are much less susceptible to virus infection due to a strong intrinsic innate antiviral immunity requiring a marked increase in the MOI used for assay set-up.

More, the hit-to-lead pipeline entails confirmation of antiviral activity in other cell lines than that used in the primary screen, including cell lines and/or primary cells naturally involved in the pathogenesis of the targeted viruses (**REF #40,57**).

We witness that there is frequent misunderstanding in the complexity of the antiviral development pipeline, whereby it is often erroneously suggested that primary hits originating from a library screen may almost directly yield a drug to treat patients. Such suggestions may include poorly characterized natural compounds, repurposed (cancer) drugs, chemical probes used in cell biological research and other rather unspecific inhibitors that do not fit the complex requirements of a state-of-the-art potent and well-tolerated antiviral medicine with a good PK/PD profile to yield therapeutic effect *in vivo* (for an earlier brief commentary on the matter by us please see PMID: 27736642); with uncountable examples until recent history, e.g., the unfortunate endorsement of anti-parasitic agents such as hydroxychloroquine or ivermectin for the treatment of COVID-19 (e.g., PMID: 32848158, PMID: 32353859, and PMID: 36915278). **To position the role of library screens in the process of drug discovery, we include a more general statements (lines 362-365) supported by new REF #40 and #57.**

Minor issues:

TBEV is not a mosquito-borne virus (lines 45-46)

Corrected to vector-borne (**line 58**).

The authors claim that the modified DENV carrying fluorescent proteins replicate similarly. However, it seems to me that they have very different fitness (the size of the plaques shown in sup figure 1 are very different). Such difference in fitness may influence the potency of the identified compounds.

Here we honestly disagree. As we previously demonstrated for DENV-2/mCherry in comparison with its parental DENV-2/WT (**REF #23**), the efficacy of directly acting antiviral compounds (measured by the percentage of inhibition in an antiviral assay) is generally not altered by the actual replication fitness of the target virus. However, we agree that the fold-reduction (\log_{10} reduction) achieved by fully inhibiting a wild-type virus that originally replicates to higher yields can be more impressive.

Please justify the use of YFV and JEV vaccine strains.

Use of vaccine strains allows to work under less stringent biosafety containment conditions (BSL2 instead of BSL3). This is important for operations, costs and feasibility of HTS. The CAPS-IT robotics platform (automated pathogen-in-operator-out high-containment lab; <https://rega.kuleuven.be/cmt/capsit>) used in our study is unique and not available elsewhere where similar screens may need to be conducted.

Moreover, use of vaccine strains for antiviral testing is well established, e.g., for YFV (**new REF #54**) and JEV (**new REF #55**), whereby major common molecular drug-targets remain unaltered compared to the respective wild-type viruses. Notably, also the mouse-adapted DV2 NGC strain (PMID: 14903434) is classified as BSL2 pathogen and was originally explored as vaccine strain.

Appreciated this concern a narrative is added to the Discussion (**lines 330-338**).

What about fitness of FP-YFV and FP-JEV as compared to unmodified counterpart?

The performance of the respective YFV/mCherry and JEV/eGFP has now been compared to their respective parental YFV17D and JEV-14-14-2 wild-type viruses (**new Supplementary Fig. 3**).

To that end, plaque assays and viruses yield assays were performed. These assays reveal consistent reduction in plaque size, yet minimal impact on final virus yields at day 7 (**new Supplementary Fig. 3**). Both findings are now mentioned in **lines 117-122** and the Extended Data file (**lines 986-998**) where the characterization of these reporter constructs are described in detail. See also discussion of added value of reduced CPE for HCI-based screening (**Lines 151-152 and 333-335**).

Line 32: please add references for HIV/HCV antiviral drugs (reviews will do)

Appropriate references have been added **as new REF #1-5**.

The 3 selected Flaviviruses are related mosquito-borne viruses. Results may not be extrapolated to more distant flaviviruses, such as TBEV for instance.

This lack in representation of tick-borne flaviviruses was added to line 314 as kind of outlook for future application of the multiplex assay principle.

Fig 1F. Please indicate the MOI and the time at which the experiment was done.

Representative photographs of virus lysis plaques were taken from titration of working virus stocks. This is now indicated in the legend to respective **figure panel 1f (lines 768-769)**.

Typos:

Line 161: NS4B instead of N4B

Corrected.

REVIEWERS' COMMENTS

Reviewer #1 (Remarks to the Author):

revision of the manuscript is adequate

Reviewer #2 (Remarks to the Author):

The authors have made extensive changes in response to my critiques. I have no additional comment.

There is a minor suggestion to include the biological or technical repeats in new Fig 4 and other places where the pie chart and scale bars are without error bars.

Reviewer #3 (Remarks to the Author):

This revised manuscript has addressed all the comments raised in the previous submission. We especially appreciate the efforts the authors have undertaken to conduct additional experiments, including a pilot screen and other multiplex infections.